# Quantifying the Influence of the Terrestrial Biosphere on Glacial-interglacial Climate Dynamics

Taraka Davies-Barnard[1,2], Andy Ridgwell[1,3], Joy Singarayer[4], and Paul Valdes[1]

[1]BRIDGE, Cabot Institute, and School of Geographical Sciences, University of Bristol, Bristol, BS8 1SS, UK
[2]College of Engineering, Mathematics, and Physical Sciences, University of Exeter, Exeter, EX4 4QE, UK
[3]Department of Earth Sciences, University of California, Riverside, CA 92521, USA
[4]Department of Meteorology and Centre for Past Climate Change, University of Reading, PO Box 243, Whiteknights Campus, Reading, RG6 6BB, UK

*Correspondence to:* Taraka Davies-Barnard (t.davies-barnard@bristol.ac.uk)

**Abstract.** The terrestrial biosphere is thought to be a key component in the climatic variability seen in the paleo record. It has a direct impact on surface temperature through changes in surface albedo and evapotranspiration (so called biogeophysical effects) and in addition, has an important indirect effect through changes in vegetation and soil carbon storage (biogeochemical effects) and hence modulates the concentrations of greenhouse gases in the atmosphere. The biogeochemical and biogeophysi-
cal effects generally have opposite signs meaning that the terrestrial biosphere could potentially have played only a very minor role in the dynamics of the glacial-interglacial cycles of the late Quaternary. Here we use a fully-coupled dynamic atmosphere-ocean-vegetation General Circulation Model (GCM) to generate a set of 62 equilibrium simulations spanning the last 120 ka. The analysis of these simulations elucidates the relative importance of the biogeophysical versus biogeochemical terrestrial biosphere interactions with climate. We find that the biogeophysical effects of vegetation account for up to an additional -0.91°C global mean cooling, with regional cooling as large as -5°C, but with considerable variability across the glacial-interglacial cycle. By comparison, while opposite in sign, our model estimates of the biogeochemical impacts are substantially smaller in magnitude. Offline simulations show a maximum of +0.33°C warming due to an increase of 25 ppm above our (pre-industrial) baseline atmospheric $CO_2$ mixing ratio. In contrast to shorter (century) time-scale projections of future terrestrial biosphere response where direct and indirect responses may at times, cancel out, we find that the biogeophysical effects consistently and strongly dominate the biogeochemical effect over the inter-glacial cycle. In addition, depending on the assumptions about soil carbon under ice-sheets and sea level rise, we find a range in terrestrial carbon storage change from a reduction in LGM carbon storage of -440 PgC, to a gain of +37 PgC, though we consider the negative part of the range more likely. We suggest that prevailing uncertainties allow for only a small net transfer of carbon between terrestrial biosphere and ocean/atmosphere implying that explaining the observed $CO_2$ ice core record could be rather simpler than previously thought.

## 1 Introduction

Terrestrial vegetation interacts with the climate in complex ways, both responding to and impacting climate conditions and hence creating an important feedback in the Earth system (e.g., Claussen, 2009; Davies-Barnard et al., 2014b; Harrison and

Prentice, 2003; Jahn et al., 2005; Matthews et al., 2003; Pongratz et al., 2010). The influence of the terrestrial biosphere on climate occurs in two distinct ways. Firstly, there are a number of biogeophysical mechanisms such as changes in albedo or evapotranspiration that provide a direct physical influence on surface climate via changes in net solar radiation transfer, infrared loss, roughness length, latent heat loss, and less directly, via changes in moisture exchange and hence transport.

Climate feedbacks driven by these changes in terrestrial vegetation have been hypothesised to be partially responsible for some of the major past climate states (e.g., Bradshaw et al., 2015; Claussen et al., 2006; Crucifix and Loutre, 2002; de Noblet et al., 1996; Zhou et al., 2012), with many studies particularly focussing on the biogeophysical effects at the last glacial maximum (LGM) (e.g., Hopcroft and Valdes, 2014; Jahn et al., 2005; Kageyama et al., 2012; O'ishi and Abe-Ouchi, 2013). The second way in which the terrestrial biosphere can influence climate is via variations in the carbon stored in vegetation and soil. This

is a crucial component for understanding changes in the carbon cycle through the last glacial-interglacial cycle (Montenegro et al., 2006) and numerous attempts have been made to estimate the total carbon storage using a range of methods, such as inferences from marine and terrestrial carbon isotopes (e.g., Shackleton et al., 1977; Bird et al., 1994), databases of pollen (e.g., Adams and Faure, 1998; Crowley, 1995), and simple and complex modelling (e.g., Prentice et al., 1993; Kaplan et al., 2002; Kähler and Fischer, 2004; Brovkin et al., 2012; O'ishi and Abe-Ouchi, 2013). The resulting range of carbon storage change

estimates is from a few hundred to about 1000 PgC (Ciais et al., 2012). One could add to this changes in the weathering of soil minerals and hence $CO_2$ uptake from the atmosphere, and nutrient, particularly phosphate, supply to the ocean and hence changes in in the ocean productivity. For simplicity, we will not address these further here (except to include a basic silicate weathering feedback in our model analysis of the impacts of terrestrial carbon storage change).

Simulations of future vegetation changes show that the biogeochemical aspect can globally be around the same magnitude

as the biogeophysical effects (e.g., Davies-Barnard et al., 2014b) meaning that there is uncertainty even in the sign of the net feedback with climate change. Both biogeophysical and biogeochemical effects likely also play an important role in past climate change and potentially the same fundamental uncertainty in the sign of the climate feedback might arise. However, model simulations have generally focussed on either the biogeophysical impacts of vegetation changes (e.g., Bradshaw et al., 2015; Claussen et al., 2006; Jahn et al., 2005; O'ishi and Abe-Ouchi, 2013; Shellito and Sloan, 2006) or biogeochemical

impacts (e.g., Kaplan et al., 2002; Ciais et al., 2012) and the question of the overall feedback on climate rarely addressed, although Claussen (2009) argues that the net effect at the LGM is dominated by the biogeophysical effects.

One of the few examples where both have been combined and the net effects of vegetation on past climate estimated over long time periods, is Brovkin et al. (2012). They used an earth system model of intermediate complexity (EMIC) to suggest that the net effect of vegetation is to decrease global temperatures during the last glacial-interglacial cycle. But the model used

is relatively coarse in resolution ($10°$ in latitude and $51°$ in longitude) and reduced in physical process complexity. This may be important because of the local and spatially heterogeneous nature of biogeophysical effects and depending on the location of the forest, the biogeophysical and biogeochemical effects of forest change can be very different (Bonan, 2008). For this reason, fully coupled General Circulation Models (GCM) are commonly used in quantifying future climate changes to vegetated land surface (Brovkin et al., 2013a, b; Davies-Barnard et al., 2015; Davin and de Noblet-Ducoudré, 2010). The importance

of considering both biogeophysical and carbon cycle impacts together at finer scale when assessing the climate impacts of

vegetation is illustrated by work quantifying the climate impacts of forest changes. Studies have found that deforestation would cause local high latitude cooling (Betts, 2000), global warming (Davin and de Noblet-Ducoudré, 2010), or even slight global cooling (Davies-Barnard et al., 2014b, a). These outcomes are not predictable from looking at the biogeophysics or terrestrial biogeochemistry alone at coarse resolutions.

Here we present the first model analysis using a fully-coupled dynamic atmosphere-ocean-vegetation GCM over the last 120 ka that quantifies the net effect of vegetation on climate. (A prior study – Singarayer and Valdes (2010) – did not have dynamic vegetation and hence could not directly evaluate the biogeophysical effects.) We separate the biogeophysical and biogeochemical effects of vegetation to understand the overall climate effect of vegetation over the last glacial cycle. We show that over the whole period the biogeophysical is the dominant effect, and that the biogeochemical impacts may have a lower

possible range than typically estimated. We also highlight how the temporal scale affects the net impact of terrestrial biosphere changes.

## 2    Methods

We use the GCM HadCM3 to run a series of simulations with and without dynamic vegetation to provide the biogeophysical changes and the land carbon changes. To look at the climate impact of those vegetation carbon changes, we then use the GCM

terrestrial carbon changes as an input to the EMIC cGENIE to calculate the resulting change in atmospheric $CO_2$ and global temperature.

For future climate changes studies, the response of atmospheric $CO_2$ concentrations (and hence climate) to changes in terrestrial carbon storage can be calculated using the Transient Response to Cumulative Emissions (TRCE) approach (Gillett et al., 2013), which demonstrated proportionality between carbon emissions and temperature rise (Goodwin et al., 2015). We

include these estimates, for completeness. However, this approach is only valid for relatively rapid changes. On the longer time scales of glacial-interglacial change, we need to take into account the full changes in ocean carbon chemistry and including the interactions of ocean and atmosphere with the solid Earth (e.g. weathering). To do this, we employ the 'cGENIE' Earth system Model of Intermediate Complexity (EMIC).

### 2.1    Climate Model Description

The GCM used in the simulations in this study is the UK Met Office Hadley Centre's HadCM3B-M2.1a and HadCM3B-M2.1aD (Valdes et al., 2017). Though not from the latest generation of climate models, HadCM3 remains an extensively used model for many research applications around the world due to its computational efficiency, which means that long integrations and many ensemble members can be run.

HadCM3 is a three dimensional, fully coupled, fully dynamic ocean, non-flux adjusted global climate model (Collins et al.,

2001). The atmosphere component, HadAM3, has a cartesian grid with a horizontal resolution of $2.5°$ x $3.75°$, 19 vertical levels and a time step of 30 minutes (Pope et al., 2000). The ocean and sea-ice component has the same horizontal resolution as the atmosphere, with 20 vertical ocean levels.

The land surface scheme used for the atmosphere component of HadCM3 is the Met Office Surface Exchange Scheme, MOSES2.1 (Gregory et al., 1994; Cox et al., 1999). MOSES can also use an additional vegetation and terrestrial carbon model, TRIFFID (Top-down Representation of Interactive Foliage and Flora Including Dynamics) (Cox, 2001; Cox et al., 1998). TRIFFID predicts the vegetation based on plant functional types using a competitive, hierarchical model. TRIFFID has two modes, equilibrium mode, which quickly brings the vegetation cover into equilibrium by running fifty years of TRIFFID for each five years of the climate model run, and dynamic, which runs TRIFFID every ten days. TRIFFID and MOSES have nine land surface types, five of which are vegetation: broadleaf trees, needle leaf trees, shrubs, $C_3$ grasses and $C_4$ grasses. These are known as plant functional types (PFTs) and have different leaf area index limits and other phenological differences in the model. Soil moisture in the model is represented on 4 layers of thicknesses (measured from the top) of 0.1, 0.25, 0.65 and 2.5 m (Essery et al., 2001).

The soil carbon is a single pool, increased by litterfall and decreased by respiration (Cox, 2001; Cox et al., 1998). The soil respiration is controlled by moisture and temperature and returns carbon dioxide to the atmosphere unless, as is the case here, the atmospheric carbon dioxide is fixed. The litterfall is an area-weight sum of the litterfall of the five PFTs in each gridcell (Cox, 2001; Cox et al., 1998). There is no permafrost component in the model, and soil in frozen regions is treated the same as in any other.

Assessment of the PI vegetation cover of HadCM3 by Valdes et al. (2017) shows good agreement with reconstructions of 1800 vegetation.

### 2.1.1 GCM Simulations and Experimental Methodology

The simulations used here are revised versions of those described in Singarayer and Valdes (2010), who used HadCM3 version HadCM3-M1, which has an older surface scheme (MOSES1) than the MOSES2.1 used here, and no dynamic vegetation. Two sets of 62 simulations were performed, covering the time period 120 - 0 ka BP:

 – The first set of 62 simulations used TRIFFID to predict vegetation changes. Each individual simulation was initialised from the previous MOSES1 simulations (which were run for 600 years) and were then run for a further 300 years with 'equilibrium' TRIFFID and a final 300 years with fully dynamic vegetation. This set will be referred to as the Dynamic set.

 – A second set of simulations uses static vegetation based on the pre-industrial simulation of the dynamic set (extrapolated to new land areas using a simple nearest neighbour algorithm). They are otherwise identical to the Dynamic set (see details below). These will be referred to as the Static set.

The differences between Dynamic and Static allows us to evaluate the biogeophysical and biogeochemical responses of terrestrial carbon cycle change.

Both sets of simulations are forced with the same changes in orbit, greenhouse gases ($CO_2$, $CH_4$, and $N_2O$) and ice sheets, as in Singarayer and Valdes (2010) except that we use a revised ice sheets extent and elevation, as discussed in Singarayer et al. (2011).

We have also added a parameterisation of water transport from ocean to ice sheet in order to ensure that ocean salinity is conserved during each simulation. In the normal configuration of HadCM3, salinity is conserved by the numerical scheme but water that accumulates as snow on ice sheets is not interactively considered. A predefined (spatially varying) flux of water is prescribed into the model which minimises the salinity drift for the pre-industrial simulation but this is not normally changed for other time periods. In our new parameterisation, we continue to add the predefined flux but also add an additional flux which is spatially uniform but temporally variable to ensure that the volume integral ocean salinity is relaxed back to its initial value, with a relaxation time scale of 10 years. This prevents any spurious long term drifts in ocean salinity.

Note that this model does not have a closed carbon cycle. There is no representation of carbon in the ocean and terrestrial carbon changes do not feedback to the atmosphere (since the greenhouse gas forcings are prescribed). However, the carbon that would have returned to the atmosphere can be inferred from the change in the carbon stores in the soil and vegetation, allowing the biogeochemical impact of vegetation to be understood, as well as the biogeophysical. From the 22 ka to pre-industrial, simulations are run for every 1000 years. From 80 ka to 22 ka, simulations are run for every 2000 years. For 120 ka to 80 ka, simulations are run for every 4000 years. (See grey points in Figure 2 for a representation of the temporal distribution of the 62 simulations.) Reported final climatologies are based on the last 30 years of each simulation.

## 2.2   EMIC Description

The cGENIE Earth system model is used to calculate the impacts on atmospheric $CO_2$ over the glacial cycle and hence make a time-varying estimate of the contribution of biogeochemical changes to glacial-interglacial climate change. The model is based around a fast energy-balance based atmosphere model coupled to a 3D ocean circulation component and dynamic-thermodynamic sea-ice (Edwards and Marsh, 2005), plus representations of ocean-atmosphere (Ridgwell and Hargreaves, 2007), ocean-sediment (Ridgwell et al., 2007), and atmosphere-land (terrestrial weathering) (Colbourn et al., 2013) carbon cycling. As employed here: the non-seasonally forced ocean has 8-levels and the configuration and selection of model parameterisations and parameter values is identical to that described in Lord et al. (2016). These choices are made to minimise experiment run-time and provide maximum traceability (to a previously used and in-depth analysed configuration), respectively.

## 2.3   cGENIE Carbon Cycle Simulations

The evolution of terrestrial carbon storage simulated by HadCM3 from 120 ka to pre-industrial was used to derive a forcing for cGENIE. In this, we created a continuous time-series of the carbon flux from the terrestrial biosphere by calculating the difference in carbon storage calculated at the end of each HadCM3 time-slice and then assuming a linear interpolation between these points. For the 'Full' simulations, cGENIE was then run for 120 ka using this forcing and starting from a fully spun-up state of global carbon cycling including an initial balance between the rate of silicate rock weathering and volcanic $CO_2$ outgassing (see Lord et al. (2016) for details). For the 'Carbonate' simulations, the model was run with just carbonate compensation only, as per Ridgwell and Hargreaves (2007); For the 'Closed' simulations, there was no weathering or sediment response, and hence is just ocean-atmosphere repartitioning. For the 'AirOcean' simulations, the carbon remains

in the atmosphere. Both the resulting history of atmospheric $CO_2$ as well as annual mean global surface air temperature were extracted and calculated as anomalies relative to the late Holocene (pre-industrial).

Using the 'Full' setup, cGENIE simulations were run using four different carbon estimations from the GCM simulations (see Table 1). For 'Carbonate', 'AirOcean' and 'Closed' a simulation was run with the GCI_ELE carbon scenario (see Table 1). Therefore seven transient cGENIE simulations were run in total.

It should be noted that we do not attempt to change the boundary conditions required by the cGENIE model dynamically through the glacial-interglacial cycles, namely: orbital parameters, planetary albedo, sea-level (and ocean salinity). These are instead kept fixed at modern (following Lord et al. (2016).) Hence, changes in the sensitivity of atmospheric $CO_2$ to unit $CO_2$ input (or removal) will not be accounted for. We expect such an effect to exist due to e.g. the dependence of the Revelle factor (the sensitivity of dissolved $CO_{2(aq)}$ to changes in total dissolved inorganic carbon (Zeebe et al., 1999)) on both (ocean surface) temperature and atmospheric $pCO_2$, changes in ocean circulation and the efficiency of the biological pump, and changes in the carbonate buffering of ocean chemistry. Some of these factors could in theory be imposed (e.g. changes in ocean surface temperatures), but others would require the glacial-interglacial dynamics in both ocean circulation and marine carbon cycling to be sufficiently accurately represented in the model. The latter is far beyond what the current state of understanding of glacial-interglacial global carbon cycling allows for (Kohfeld and Ridgwell, 2009). Hence our assumption of fixed late Holocene boundary conditions will impart a small bias in our estimates of the atmospheric $CO_2$ response, but not one that would affect our overall conclusions.

In addition, in making estimates of the mean global air surface temperature change corresponding to the projected change in atmospheric $pCO_2$ in cGENIE, it is important to also note that the climate sensitivity is effectively prescribed (Edwards and Marsh, 2005). In the Lord et al. (2016) configured used here, only sea-ice cover, via its associated albedo, can provide feedback on climate. In the absence of a dynamical atmosphere, glacial-interglacial changes in climate sensitivity due to changes in atmospheric circulation and clouds are not possible. Nor do we account for the possible influences of changes in total land surface area (from sea-level change) or vegetation cover and distribution. However, the assumption of an effectively fixed climate sensitivity across the glacial-interglacial cycle is unlikely to impart significant bias or unduly affect our overall conclusions.

## 3 Results

### 3.1 Results: Vegetation Dynamics

The changes in climate over time affects the vegetation cover in the Dynamic simulations (shown in Figure 1). In general, cooling leads to an equator-ward shift in vegetation, as the high latitudes become covered in ice or otherwise inhospitable for significant quantities of vegetation. There is also exposure of continental shelves, providing potential for vegetation increases. At the last glacial maximum (LGM) at 21 ka, we can see needleleaf trees and shrubs giving way to very low productivity grasses in the high latitudes. However, because of the small number of PFTs (five) in this model, the shifts may be underestimated, as each PFT represents a wide range of vegetation types. The shrubs and trees do not have a significant presence in northern

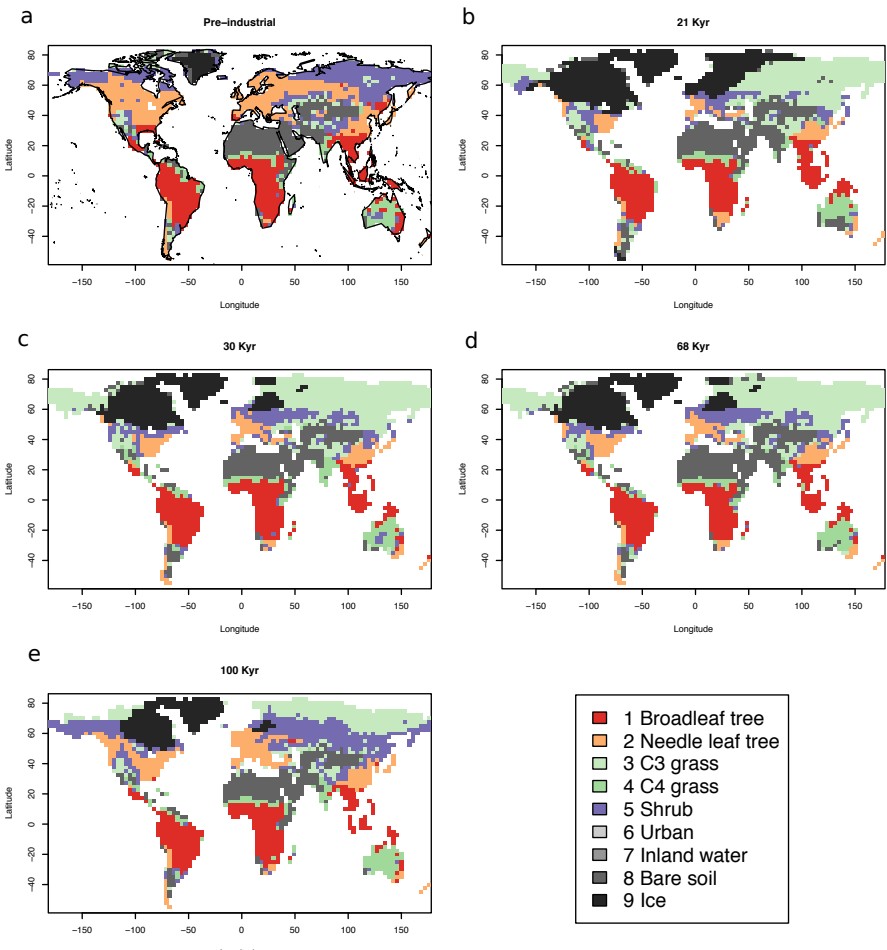

**Figure 1.** Dominant plant functional type (PFT) for some time periods of interest in the 120 ka covered by the simulations. a) Pre-Industrial, b) 21 ka, c) 30 ka d) 68 ka, e) 100 ka. Note that the dominant PFT is calculated as the land cover with the highest proportion of cover, compared to the other land surface types, and does not necessarily indicate the highest or a significant amount of net primary productivity (NPP).

Europe after 100 ka until the climate ameliorates into the Holocene. It is the vegetation changes shown in Figure 1, and their associated soil changes, that drive the climate feedbacks and other changes described hereafter.

Hoogakker et al. (2016) have shown that HadCM3 broadly reproduces the known changes in vegetation across the glacial-interglacial cycle. Hoogakker et al. (2016) uses HadCM3B-M1 (without dynamic vegetation), then uses the climate to drive
5 BIOME4. The climate is very similar between HadCM3B-M1 and HadCM3B-M2.1a used here. In Hoogakker et al. (2016) they ran an offline vegetation model, BIOME4, driven by the climate anomalies from HadCM3. Our results from TRIFFID are consistent with the relative changes although, since TRIFFID uses the actual climate from the models, the vegetation can

have biases (e.g. Australia has a tendency to be too wet in HadCM3 in the present day and hence the coupled model has too much vegetation in this region). However, during glacial times there is a decrease in biomass, consistent with Hoogakker et al. (2016).

Comparison with the BIOME6000 Mega-Biome maps for LGM (Pickett et al., 2004; Prentice and Jolly, 2000; Bigelow et al., 2003; Harrison et al., 2001) dataset shows general agreement. The model has considerable expansion of grasses in Eurasia where BIOME6000 has grassland and dry shrubland. Broadly speaking, in both North America shows little change from the mid Holocene to LGM. One key weakness of the model is in western Europe, where BIOME6000 shows grassland and dry shrubland, whereas the model has shrubs and needleleaf trees. South-east Asia shows continued Warm-temperate, Temperate, and Tropical forest where our model simulated Broadleaf trees, which encompasses all of these biomes. The BIOME6000 reconstructions show around a dozen Tundra points on and near the bering land bridge, and our model simulates this as $C_3$ grasses, which is the closest PFT to Tundra. Over central Asia our model has extensive areas where the dominant land surface type is bare soil, indicating desert or sparse, dry vegetation. BIOME6000 shows a mixture of desert and dry grass/shrubland, which is generally in keeping with the low productivity, low density vegetation indicated in our model simulation.

The forest extent in the tropics at the LGM is similar to PI (see SI Figure 9 for shifts in vegetation at 21 ka). This is supported by pollen and other data (Maslin et al., 2012; Anhuf et al., 2006; Wang et al., 2017), and modelling (Cowling et al., 2001) which find that although there is diminished tropical forest, there is still substantial tree cover at the LGM and little sign of widespread grasslands. Conversely, the BIOME6000 data finds that the tropical rainforest area was reduced during the LGM (Pickett et al., 2004; Prentice and Jolly, 2000; Bigelow et al., 2003; Harrison et al., 2001) and grasslands expanded, as do some modelling studies (Martin Calvo and Prentice, 2015; Prentice et al., 2011; Hoogakker et al., 2016). It is interesting to note that in the present day Amazon, BIOME6000 shows 3 points of tropical forest; 2 Savanna, 2 Warm-temperature forest; 2 temperate forest; and 3 dry grass/shrubland at the LGM. In our simulations the dominant PFT of the same area is broadleaf-trees. For comparison, Prentice et al. (2011) using LPX have tropical forest over the same domain. Therefore there is little indication that where TRIFFID may be inconsistent with BIOME6000 that another model is necessarily significantly better.

Because of the PFT (rather than biome) approach of TRIFFID, and the limited number of PFTs, it's difficult to be sure whether trees in the tropics are a tropical rainforest at the LGM, as there are a number of biomes with significant amounts of trees. Although there is little change in PFT in the tropics at the LGM, on the margins there are reductions of vegetation carbon, suggesting a change in vegetation within the large margins of the PFTs used in this model.

## 3.2   Results: Biogeophysical Feedbacks

The biogeophysical impacts of vegetation are calculated by subtracting the Dynamic simulations from the corresponding Static simulations. We find that vegetation is acting as a positive feedback to the climate, enhancing the cooling (Figure 2a). Broadly, the Static and Dynamic simulations both agree with an approximation of global temperature over the whole period (the EPICA dataset halved) (Figure 2a). The Static set generally do better in 70 ka to 10 ka, whereas the Dynamic set are closer to the EPICA data in the period 110 to 70 ka. The biogeophysical differences between the Static and Dynamic sets alter global,

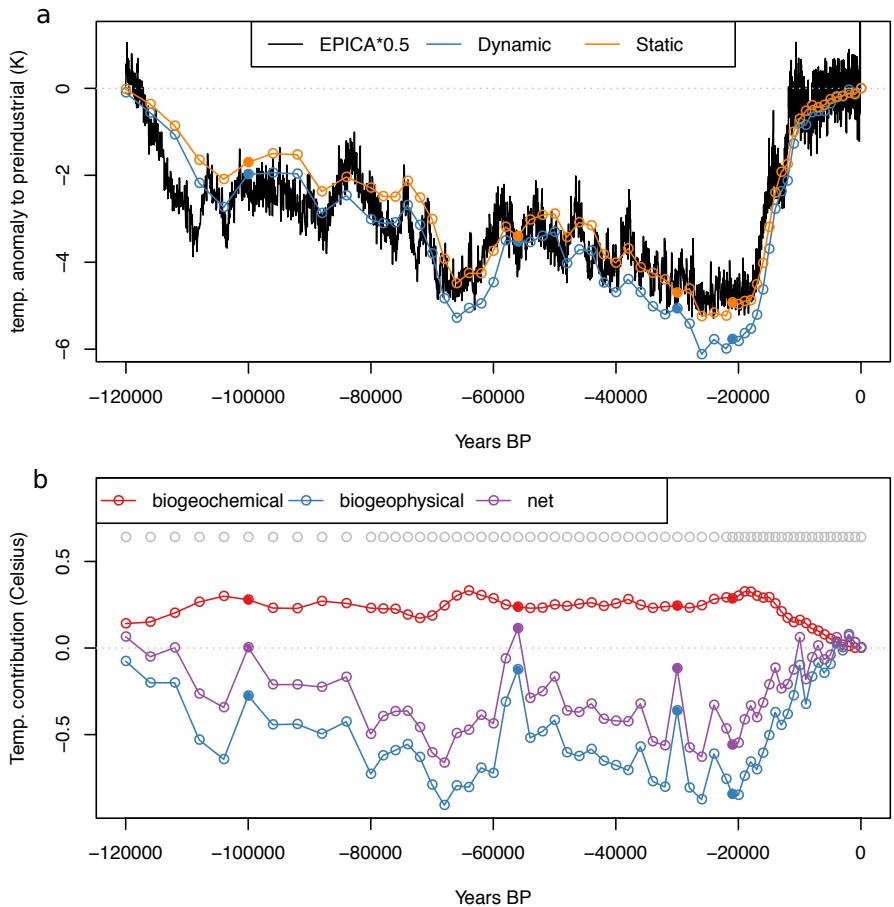

**Figure 2.** a) Global mean annual temperature (Celsius at 1.5m) for: Static vegetation simulation (orange); Dynamic vegetation (blue); and the EPICA core inferred temperature data (black), halved (to give an approximation of global temperature over the whole period). Time periods of particular interest are highlighted as filled points: 21, 30, 56, 68 and 100 ka. b) Temperature anomaly over time of Dynamic - Static simulations for: biogeochemical temperature effects of the vegetation change, calculated with GENIE, averaged to the same temporal resolution as the HadCM3 simulations (red); biogeophysical temperature effects of vegetation change (blue); the net (biogeophysical and biogeochemical) effect of vegetation on temperature (purple). Grey points show the time points of the HadCM3 simulations.

annual mean surface temperature by as much as -0.91°C (see Figure 2b). Regionally this temperature cooling is up to 5°C (Figure 3).

The albedo changes are in the same location as the vegetation carbon changes, and the main temperature changes (see Figure 4 and 3). These temperature differences are mainly driven by reductions of tree cover and its replacement with bare soil or
5  grasses, which is a result of the vegetation dynamics in the model (see Figure 1 and Figure 4). Trees have a lower albedo, and when they are replaced by higher albedo grasses, there is a cooling effect. The change in tree fraction between the Static and Dynamic sets is a good predictor of the temperature changes ($r^2 = 0.79$ using a linear model of the global temperature and tree

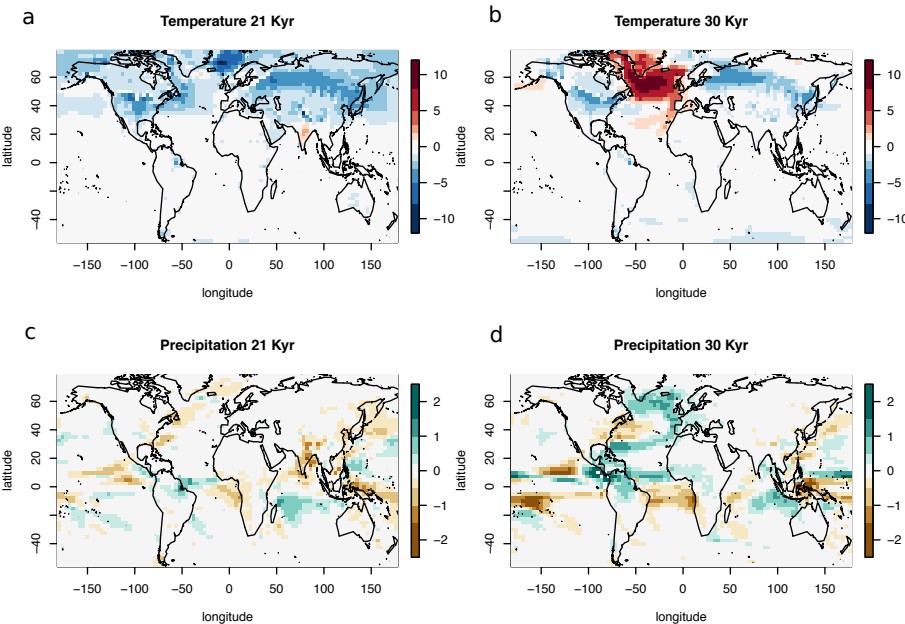

**Figure 3.** Anomaly of (a and b) temperature at 1.5m (Celsius); and (c and d) precipitation (mm day$^{-1}$); between the Dynamic Vegetation simulation and the equivalent Static Vegetation simulation. For (a and c) 21 ka and (b and d) 30 ka. The pattern of reduced temperature at 30 ka is similar to the pattern at 56 ka and 100 ka.

anomalies). This is exacerbated by the presence of snow cover as the snow covered visible and near infrared albedo of grasses, shrubs and bare soil is higher than that of trees (Essery et al., 2001). Therefore when trees are replaced by grasses where there is snow cover for part of the year, there is a larger change in albedo than where there is no snow cover. Thus the albedo changes can be seen mainly where a change between trees and grasses occurs in an area with snow cover (see Figure 1 and Figure 5).

The exact contribution of the snow as opposed to the no-snow albedo is difficult to disentangle, but the influence of this effect is well established (Betts, 2000).

The land surface albedo changes caused by the vegetation have an even stronger correlation with these biogeophysical temperature changes ($r^2$ = 0.86). However, we can see that although the forcing is land based (the dynamic vegetation), significant changes occur in the ocean (see Figure 4 and Figure 3) that drive the resulting temperature changes. Ocean only

10 surface albedo anomaly as a determinant of global temperature anomaly has an $r^2$ of 0.95 - lower only than the $r^2$ of the global (land and ocean) surface albedo of 0.96. By comparison, the $r^2$ of the latent heat anomaly as a predictor of temperature anomaly is lower for land, ocean, and globally than surface albedo (0.70, 0.93 and 0.91 respectively). The other parts of the energy balance, in particular the latent heat, sensible heat, and the net shortwave radiation, do not have such a clear relationship with the temperature change (see SI Figure 11 and compare to Figure 3).

Although the biogeophysical changes cause cooling, there are some minima of biogeophysical temperature change seen at 30 ka, 56 ka and 100 ka (Figure 2, filled symbols). These minima have an oceanic source and are caused by vegetation interacting

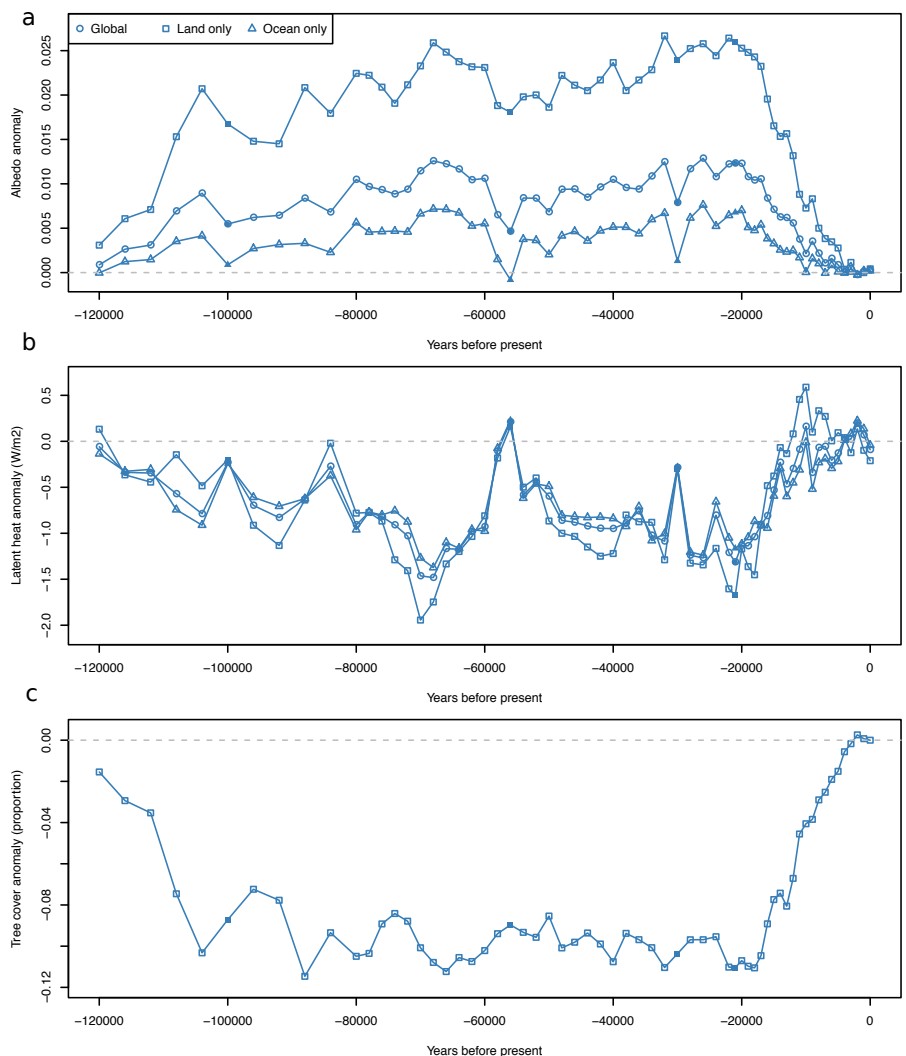

**Figure 4.** Mean annual anomaly Dynamic - Static simulations for: a) surface albedo b) latent heat (W/m$^2$) and c) tree cover (proportion of land area).

with thermohaline circulation changes. In our new simulations we account for the net transport of water from ocean to the ice sheets by a parameterisation that instantaneously balances any net accumulation of water on ice. This parameterisation results in fresher ocean conditions during times of precession driven N. Hemisphere summer insolation highs (less water is being used to build the ice sheets). The instantaneous nature of the parameterisation is physically unrealistic but reductions in accumulation and an increase in ablation during precession highs has been seen in fully coupled climate-icesheet EMIC simulations (e.g. Ganopolski et al, 2010). During weaker accumulation periods, the parameterisation results in a freshening of ocean surface waters and a reduction in AMOC strength from  16Sv to 10 -12Sv.

Superimposed upon this general behaviour, the addition of interactive vegetation generally does not change the AMOC strength. However, at times of weak AMOC, small changes in runoff and temperature are sufficient to cause some changes in the response. For instance, in the static vegetation simulations there is a relatively weak AMOC in the simulations for 60 ka, 58 ka, and 56 ka. In the interactive vegetation simulations, the weakened AMOC only occurs at 60 ka. Thus at 60 ka the changes in climate are fairly typical of preceding times but at 58 ka and 56 ka there is a substantial difference between the static and dynamic vegetation simulations. The cause for this difference is associated with a combination of reduced runoff into the N. Atlantic (principally from changes in land surface in N. America) and colder temperatures, both of which act to stabilise the AMOC in all three periods but it is sufficient to prevent the AMOC weakening in the 58 and 56 ka simulations. This threshold like behaviour of the AMOC is almost certainly highly model dependent and hence the result is not robust.

The regional patterns of cooling also temporarily affect the precipitation regime (see Figure 3). This appears to be related to the AMOC weakening. There are some suggestions of similar relationships between the increases in precipitation and the terrestrial changes to previous studies (Gedney and Valdes, 2000; Singarayer et al., 2009). Similar to the temperature changes, it is unclear how model-specific these changes are.

### 3.3 Results: Biogeochemistry

We now calculate the total change in terrestrial carbon stores in the HadCM3 simulations. We consider scenarios of terrestrial carbon change with combinations of including or excluding uncertain aspects of the carbon cycle, specifically depending on the fate of soil carbon under ice and the changes related to the expansion of land. Zeng (2003) suggested that the soil and vegetation carbon formed during the warm last interglacial could simply get covered by ice and is stored there, rather than being released into the rest of the system as is typically assumed in past estimates. Similarly, the amount of carbon stored on newly emerged land is also uncertain as it depends on both the area of emergent land and the surface properties. Therefore we calculate the changes in soil and vegetation carbon from these various sources. In Table 1 we focus on the changes between pre-industrial and LGM, which corresponds to the largest overall change through the glacial-interglacial cycle.

In the model, 222 PgC of soil carbon and 73 PgC of vegetation carbon is associated with areas covered with ice at the LGM (see Figure 8). Similarly, 134 PgC of soil carbon and 49 PgC of vegetation carbon is associated with new land. The resulting range of total carbon storage is large, from a loss of 440 PgC at the LGM (no carbon stored under new ice sheets with all being released to ocean-atmosphere, and no build-up of carbon on new land surface) to a possible small increase of carbon (if carbon is stored under the new ice sheets and there is no carbon storage on new land).

In reality, glacial systems are known to export carbon in a highly labile form (Lawson et al., 2014), erode soil and bedrock creating major landscape changes, and release large amounts of methane when they retreat (Wadham et al., 2012). Although the conversion of this terrestrial carbon to atmospheric carbon may be through riverine or oceanic systems, it seems likely it would return to the atmosphere within the time periods we consider. We therefore use this largest scenario as a conservative option for our main analysis.

The other major change to soil carbon in the model is newly exposed land, which is revealed when the water in the ice-sheets causes lower sea levels (see Figures 1 and 5). For the new land we use a nearest-neighbour interpolation of basic soil properties

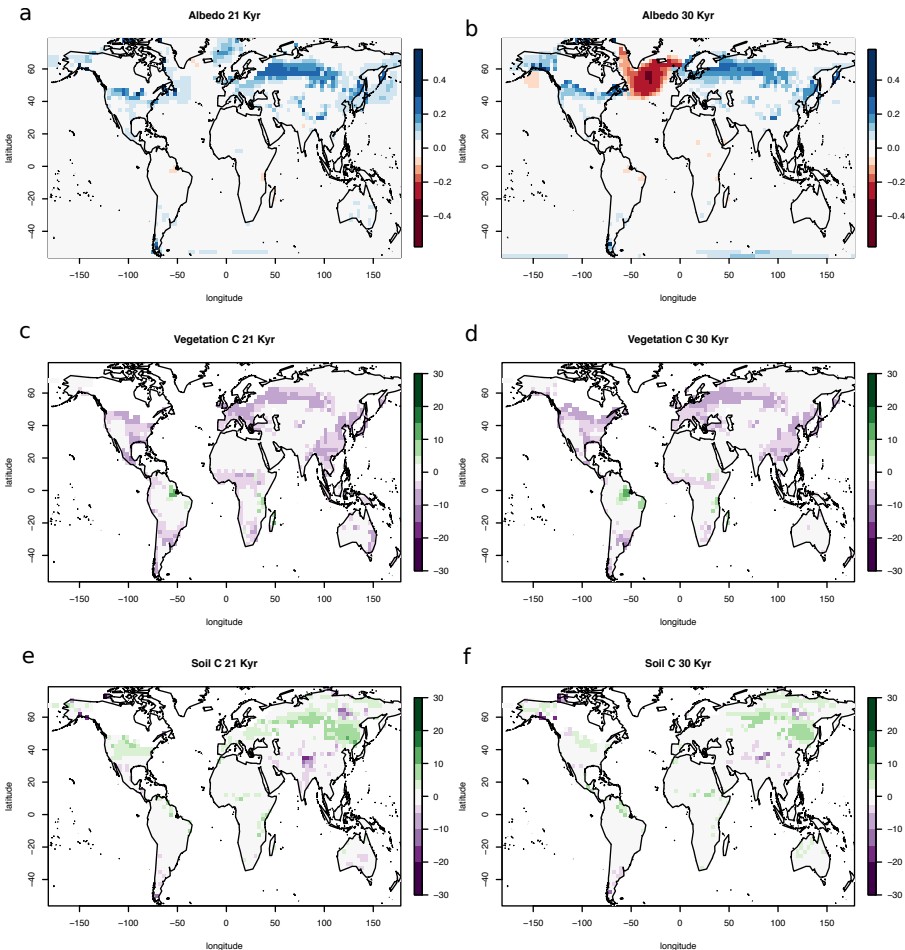

**Figure 5.** Mean annual anomaly of: a) and b), surface albedo (unitless); c) and d), vegetation carbon (kg C m$^{-2}$); and e) and f), soil carbon (kg C m$^{-2}$), between the Dynamic Vegetation simulation and the equivalent Static Vegetation simulation. For 21 ka (a, c, and e) and 30 ka (b, d, and f). The pattern of reduced surface albedo at 30 ka is similar to the pattern at 56 ka and 100 ka. Since in the Static simulations the carbon remains at PI levels, figures c - f also represent the anomaly to PI. The ice-sheets are excluded from these plots. For the carbon under ice-sheets, see Figure 11.

(e.g. water holding capacity etc.) and the model is run for sufficient length of time for the soil and vegetation carbon to reach equilibrium.

This estimate of carbon on expanded lands also has uncertainties. We have reasonable confidence in the sea level estimates and consequent change in land area, but it is much more uncertain whether carbon could accumulate on that land. For instance, in our simulations the East Siberia ice-sheet is absent (see Figure 1), whereas many other ice-sheet reconstructions include it (e.g., Niessen et al., 2013). The area of the ice sheet alone accounts for an average of 56 PgC soil carbon in these simulations

**Table 1.** Terrestrial carbon changes in the Dynamic simulations from PI to LGM. These numbers are the global totals of the maps in Figure 11 c and e, with the specified parts included/excluded. For the vegetation carbon, these are the values associated with the vegetation changes shown in 10. For storage values at the LGM, see Table A1.

| Name | Carbon storage Scenarios | Soil C change (PgC) | Vegetation C change (PgC) | Total C change (PgC) |
|------|--------------------------|---------------------|---------------------------|----------------------|
| GCI_ELE | Carbon under ice sheets released to atmosphere. No carbon on expanded land area. | -145 | -295 | -440 |
| GCE_ELE | Carbon under ice sheets stored under the ice. No carbon on expanded land area. | +77 | -222 | -146 |
| GCE_ELI | Carbon under ice sheets stored under the ice. Modelled carbon storage on new land included. | +211 | -173 | +37 |
| GCI_ELI | Carbon under ice sheets released to atmosphere. Modelled carbon storage on new land included. | -11 | -246 | -257 |

when it is absent. But soil carbon takes a long time to accumulate, especially with low NPP and vegetation carbon storage averages just 0.5 PgC over all the expanded lands.

If exposed land carbon was included and glacial land soil carbon excluded, the terrestrial carbon is a gain from PI to LGM of +37 PgC (see Table 1). However, as discussed above, we would argue that excluding glacial land soil carbon change is probably unreasonable. Most previous studies have also assumed that all carbon under ice is removed. If we include the loss of carbon, then the range in total amount of terrestrial carbon lost in this model between pre-industrial (PI) and the LGM at 21 ka is -440 to -257 PgC.

The change in terrestrial carbon found in our simulations contributes to atmospheric carbon dioxide change. Using the cGENIE model to approximate the carbon uptake by the ocean we therefore calculate the atmospheric carbon dioxide change (see methods and Figure 6).

Selecting the largest change in carbon storage (-440 PgC at the LGM, including glacial soil carbon changes and excluding expanded lands) the results suggest a peak contribution compared to pre-industrial $CO_2$ of 25 ppm $CO_2$ (Figure 6). In all scenarios except GCE_ELI, the terrestrial carbon contribution to atmospheric $CO_2$ acts as a negative feedback to the climate, dampening the effect of other climate forcings, including the net contribution of the terrestrial biosphere (Figure 2b).

Within cGENIE, the change in atmospheric $CO_2$ produces a warming at the LGM of 0.29°C (equivalent to a climate sensitivity of around 2 Wm$^{-2}$ °C$^{-1}$, see Figure 2b). This is much smaller than the biogeophysical contribution of -0.84 °C. It is also

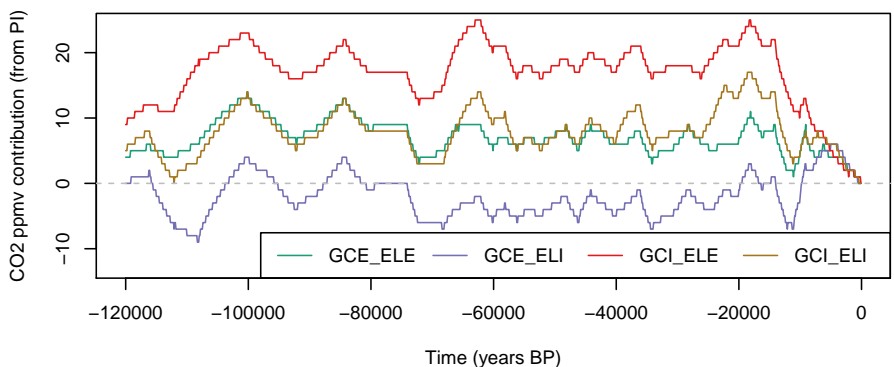

**Figure 6.** Contribution of terrestrial biosphere carbon emissions to atmospheric $CO_2$. Run with the cGENIE 'Full' configuration and normalised to pre-industrial $CO_2$ levels. The four scenarios are as detailed in Table 1.

much less variable. For most of the glacial period, from 100 ka to 20 ka, the implied biogeochemical warming is around 0.26 °C (Figure 2b). This results in the dominance of the biogeophysical impacts over biogeochemical feedbacks.

However, with different earth system processes included, the biogeochemical effects vary substantially (see Figure 7). In the simulations discussed above, silicate and carbonate weathering are both included and this results in the lowest temperature change from the same carbon emissions. The temperature contribution at the LGM increases (from Full, 0.29°C); as the silicate weathering is excluded (Carbonate, 0.30°C); all weathering is excluded (Closed, 0.47°C); a decadal to millennial scale carbon uptake is used (TRCE, 0.86°C); and if all carbon remains in the atmosphere (AirOcean, 1.92°C). Note that the TRCE as shown above includes the terrestrial biosphere as a sink, so will slightly overestimate how much carbon will be removed from the atmosphere when the source is the natural vegetation. Comparing these values to the biogeophysical terrestrial effect in Figure 2b, we can see that the shorter the timescale, the more likely biogeochemical terrestrial processes will dominate as it weakens over time. On longer timescales the biogeophysics dominates because the scale of the effect doesn't diminish over time relative to the control.

## 4 Discussion

The biogeophysical results found here broadly concur with comparable model studies of past vegetation biogeophysics. Claussen et al. (2006) found the biogeophysical contribution of vegetation to LGM cooling of around 1°C in the northern hemisphere, whereas Jahn et al. (2005) found around -0.6°C, and up to 2 °C locally. Our result of -0.84°C is in the middle of the other LGM studies.

The dominance of the biogeophysical effects found here is contrary to the results found for short time scale problems, which find that biogeochemistry tends to be comparable in magnitude to biogeophysical effects (e.g., Davies-Barnard et al., 2014b; Pongratz et al., 2010). This is because the centennial simulations have a stronger biogeochemical effect since the transient response to cumulative emissions is stronger than the equilibrium response. In climate simulations up to around a century long,

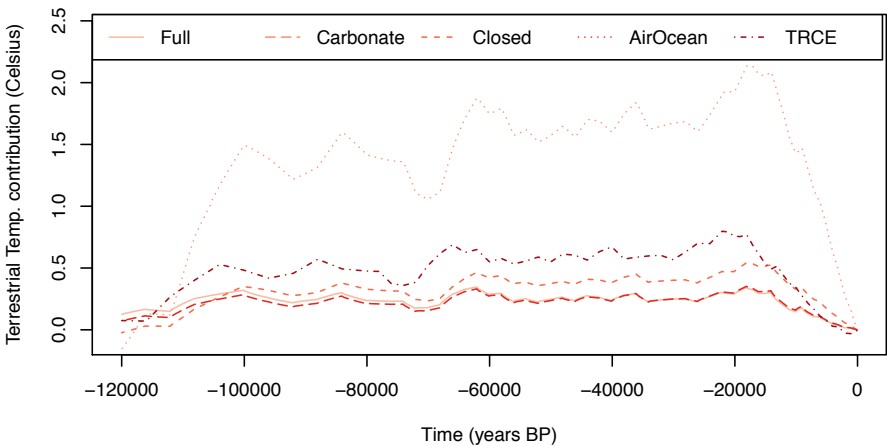

**Figure 7.** Temperature changes resulting from the same terrestrial carbon emissions scenario (GCI_ELE) with different model set-up for cGENIE and for the TRCE of HadCM3. cGENIE simulations were: 'Full' with silicate weathering feedback and just carbonate compensation, as Lord et al. (2016); 'Carbonate' with just carbonate compensation only, as Ridgwell and Hargreaves (2007); 'Closed' with no weathering or sediment response, and hence is just ocean-atmosphere repartitioning; 'AirOcean' where the carbon remains in the atmosphere. 'TRCE' is the simple calculation of the TRCE of HadCM3 (taken from Matthews et al. (2009)) for the same carbon inputs into the atmosphere as used for the cGENIE simulations. Note, we include the TRCE for completeness, but it is not a cGENIE simulation (see methods).

more carbon tends to remain in the atmosphere. This makes a strong warming effect that is approximately linearly related to the amount of greenhouse gas emissions (Matthews et al., 2009; Gillett et al., 2013). The transient response to cumulative emissions (TRCE) accounts for the uptake of atmospheric carbon by the ocean and terrestrial biosphere, but only on short timescales. The uptake of atmospheric carbon by the ocean requires hundreds or thousands of years, and is slower when the increase of carbon

into the system is small and staggered (Lord et al., 2016). However, the simulations we use are on a millennial timescale, allowing much of the carbon to be taken up by the ocean (Lord et al., 2016). From a climate sensitivity point of view, this means that on shorter timescales, the effects of dynamic vegetation can cancel each other out. This provides some rationale for the fact that dynamic vegetation has been generally not included in the majority of state-of-the-art earth system models used in CMIP5, as it doesn't significantly affect the climate sensitivity. At longer time scales, it is more important to include dynamic

vegetation, as without the positive feedback of the biogeophysical effects, the climate sensitivity would be under estimated.

For the biogeochemical effects of the terrestrial biosphere, previous estimates of carbon stocks on exposed continental shelves based on models are between 112 to 323 PgC at the LGM (Montenegro et al., 2006). The comparable number in this simulation is 183 PgC, which is on the lower end of the wide range of other models. However, it has good agreement with the vegetation reconstruction (not model) values by Montenegro et al. (2006) of 182 to 220 PgC.

The LGM terrestrial carbon change here is -440 to +37 PgC, including a zero contribution of terrestrial carbon. This is smaller than the values of -900 to -400 PgC range reviewed by Kohfeld and Ridgwell (2009). More recent modelling studies are also somewhat larger than our estimate range, such as -500 PgC (Brovkin et al., 2012), -597 PgC (O'ishi and Abe-Ouchi,

2013), and (Prentice et al., 2011) -550 to -694 PgC. However, recent inventory and isotope approaches are closer to our range of values, e.g. -378 $\pm$ 88 PgC (Menviel et al., 2017) and -330 PgC (Ciais et al., 2012).

For present day, Hugelius et al. (2014) shows around 75 - 100 kg C m$^2$ of soil carbon in far north Siberia, 20 - 40 further south. Far northern Canada is much more heterogeneous, with values from 20 - 150 kg C m$^2$. The modelled PI values are on the low side, and much more homogeneous, around 15 - 20 kg C m$^2$, but is similar to Hugelius et al. (2014) in that it shows far north America to be less consistent, with some higher areas of 35 - 40kg C m$^2$ in the far north. (See Supplementary Figure 11 of the loss of soil carbon under ice-sheets at LGM.) What this suggests is that while on the correct order of magnitude, the model has a very modest amount of soil carbon that could be considered permafrost. Therefore, we think it's reasonable to include this low estimate of soil carbon in the uncertainties.

The soil carbon change under ice-sheets between PI and LGM is modelled as ~220 PgC. Extrapolating from a comparison with Hugelius et al. (2014), this might be a third too little. If the true value were ~330 PgC, this would make the total C change PI to LGM 550 PgC, more in line with some previous model estimates. It would affect the global mean annual biogeochemical contribution by ~0.1 K. This would mean the net effect of vegetation was closer to zero, but the biogeophysical effect would still dominate.

However, the exact size of the terrestrial carbon emissions is uncertain. Other carbon stores not accounted for here are potentially important, for example methane during sea level rises or changes to the wetlands in the tropics. Modelling studies that look at wetlands at the LGM suggest that although the wetland area is larger, the methane emissions are lower compared to modern day (Kaplan, 2002). However, paleohydrological data indicates a drying in the African tropics (Gasse, 2000). Our model does not have a process based permafrost or wetlands component, and therefore the changes in methane are not accounted for. This is a particular limitation when considering the carbon stored in deep permafrost soils in Northern peatlands. Saito et al. (2013) show that, based on the temperature changes, there is a substantial expansion of permafrost area during glacial times but cannot estimate any changes in carbon storage. Zimov et al. (2006, 2009) have argued that permafrost storage could be a major source of carbon through the deglaciation, and Ciais et al. (2012) argue that there was a large extra pool of inert carbon at the LGM. Similarly, Köhler et al. (2014) have argued that large amounts of carbon were locked into permafrost which were then released rapidly at the Bolling-Allerod.

Research has also suggested that waterlogging and flooding as sea level rises during the Holocene could cause rapid anaerobic decomposition of vegetation, causing methane emissions (Ridgwell et al., 2012). This could account for emissions of as much as 25 PgC for 10 meters sea level rise (ibid). Since our simulations do not account for methane or this effect of inundation, it is likely it there is a slight underestimation of equivalent CO$_2$ effect of the carbon emissions (as methane is a stronger greenhouse gas than carbon dioxide).

The impacts are mainly determined by the vegetation shifts the DGVM simulates. Each grid-box has the potential for 5 PFTs, but generally the Lotka-Volterra equations used in TRIFFID mean that the grid-box is dominated by one PFT. The small number that means the range within each PFT is relatively large. Therefore the model probably underestimates the effects of small perturbations in climate, as the large definition of the PFTs allows the PFT to remain the same. Conversely, it makes an abrupt change more likely as the climate tips a grid-box from being predominantly one PFT to being predominantly another.

Overall, the model could be slightly underestimating the amount of change in vegetation. However, because of the ratio of the biogeophysical to biogeochemical changes, if the vegetation change is underestimated, the sign of the net effect of the terrestrial biosphere is unlikely to change. Similarly, because on the long time periods involved much of the released carbon is taken up by the ocean, the changes in carbon densities of the vegetation would need to be wrong by a lot to change the overall signal.

Our approach here assumes that there is no non-linear interaction between the biogeochemical and biogeophysical effects. Since the biogeochemistry acts as a negative feedback and reduces over time, and the biogeophysics acts as a positive feedback and stays the same over time, there's no strong reason to believe that in equilibrium there would be any significant synergy. However, on shorter timescales and on a regional rather than global scale, it is quite possible that there could be some synergies.

## 5 Conclusions

Using a fully coupled atmosphere-ocean-vegetation model with static and dynamic vegetation, we find that over the last 120 ka the net effect of vegetation feedbacks on global, annual mean 1.5m air temperature is a cooling, which can be as much as -0.66°C (Figure 2 b). For the vast majority of the last glacial-interglacial cycle, cooling associated with biogeophysical feedbacks dominate over the biogeochemical warming associated with reduced terrestrial carbon storage. The biogeophysical cooling effect is mainly due to the role that vegetation plays in changing surface albedo and particularly related to snow cover and the taiga/tundra transition (Gallimore and Kutzbach, 1996; de Noblet et al., 1996) and we believe is relatively robust. The biogeochemical contribution to atmospheric carbon dioxide is small (~20ppmv) and hence the temperature contribution is small (on average 0.26°C with a maximum of 0.33°C). There are significant uncertainties in this calculation which would further diminish the net temperature impact of the terrestrial biosphere by cancelling out the biogeophysical impact. In this analysis, the only time periods where the effects are comparable are at times when additional mechanisms operate, such as changes in ocean circulation, but these mechanisms may be model specific.

The key uncertainties in this study originate in the biogeochemistry, especially the soil carbon build-up in newly exposed land, the fate of soil carbon in glacial systems, and the amount of carbon in permafrost (not calculated in this study). Further research is needed to fully understand the functioning of these systems and how they can be best incorporated into climate models. In addition, the technique we use for inferring the biogeochemical effects of terrestrial carbon changes has limitations and is potentially model dependent. However, the smaller estimate of terrestrial carbon emissions may make the low LGM atmospheric carbon dioxide somewhat easier to reconcile (Montenegro et al., 2006).

Our work confirms previous results using EMICs Brovkin et al. (2012) that found the net terrestrial biosphere effect to be primarily biogeophysical and that the terrestrial carbon contribution to atmospheric carbon is comparatively small. Our findings also represent a clear illustration of the net climatic effect of vegetation is highly dependent on the timescale, with the biogeophysical response dominating in the longer term in contrast to century-scale future changes.

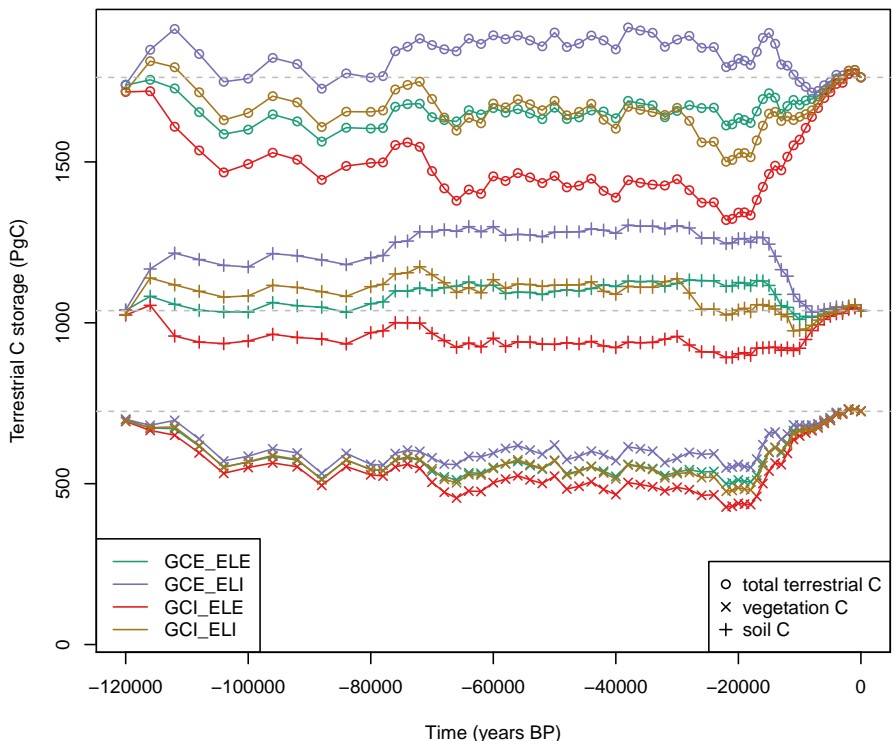

**Figure 8.** Absolute changes of carbon stores (vegetation and soil) over time. The four scenarios are as detailed in Table 1.

## 6 Code availability

The model code is currently available to view at http://cms.ncas.ac.uk/code_browsers/UM4.5/UMbrowser/index.html.

## 7 Data availability

The GCM simulation data is available at http://www.paleo.bris.ac.uk/ummodel/scripts/papers/Davies-Barnard_et_al_2017.
5 html.

## Appendix A

*Author contributions.* PJV and JSS ran the climate model simulations. TDB did the analysis and wrote the manuscript. AR ran the cGENIE model simulations. All the authors provided comments and contributed to the manuscript.

*Competing interests.* The authors have no competing interests to declare.

**Table A1.** Terrestrial carbon storage at the LGM.

| Carbon storage Scenarios | Soil C (PgC) | Vegetation C (PgC) | Total C (PgC) |
| --- | --- | --- | --- |
| GCI_ELE Carbon under ice sheets released to atmosphere. No carbon on expanded land area. | 893 | 430 | 1323 |
| GCE_ELE Carbon under ice sheets remains stored under the ice. No carbon on expanded land area. | 1114 | 502 | 1617 |
| GCE_ELI Carbon under ice sheets remains stored under the ice. Modelled carbon storage on new land included. | 1249 | 552 | 1800 |
| GCI_ELI Carbon under ice sheets released to atmosphere. Modelled carbon storage on new land included. | 1027 | 479 | 1506 |

*Acknowledgements.* T Davies-Barnard was funded by the European Research Council's grant ERC-2013-CoG-617313 (PaleoGENIE).

This work was carried out using the computational facilities of the Advanced Computing Research Centre, University of Bristol - http://www.bris.ac.uk/acrc/.

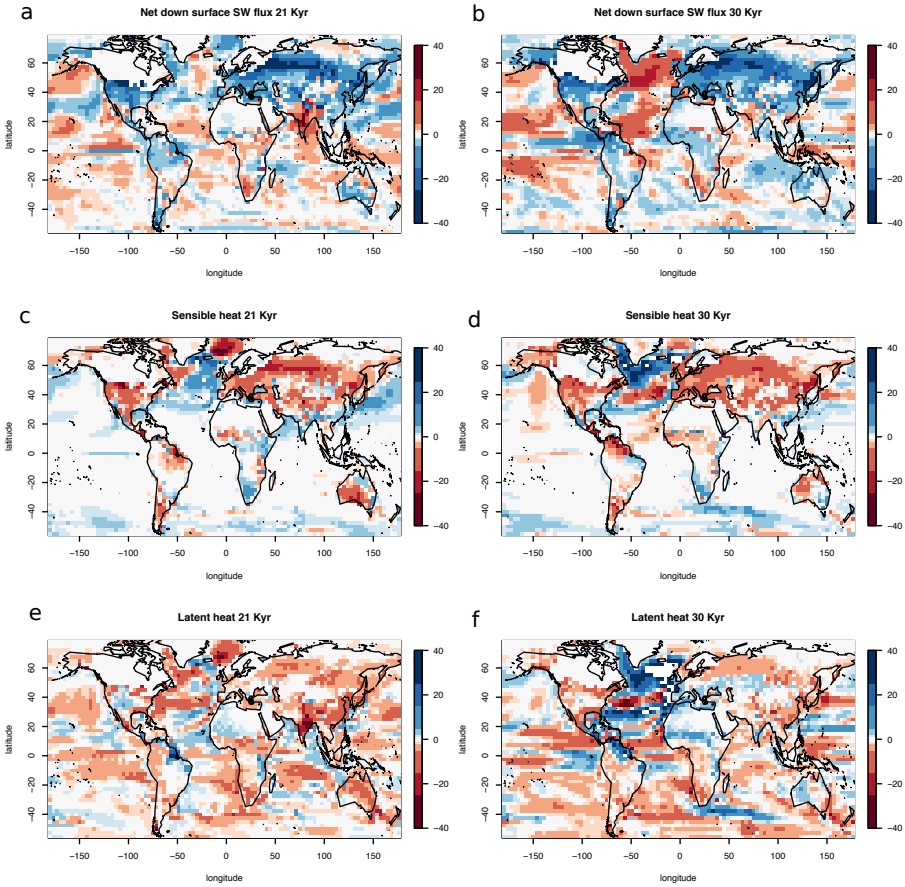

**Figure 9.** Maps of mean annual anomalies between Dynamic and Static simulations for net down short wave flux, sensible heat, and latent heat, for the 21 kyr and 30 kyr simulations.

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

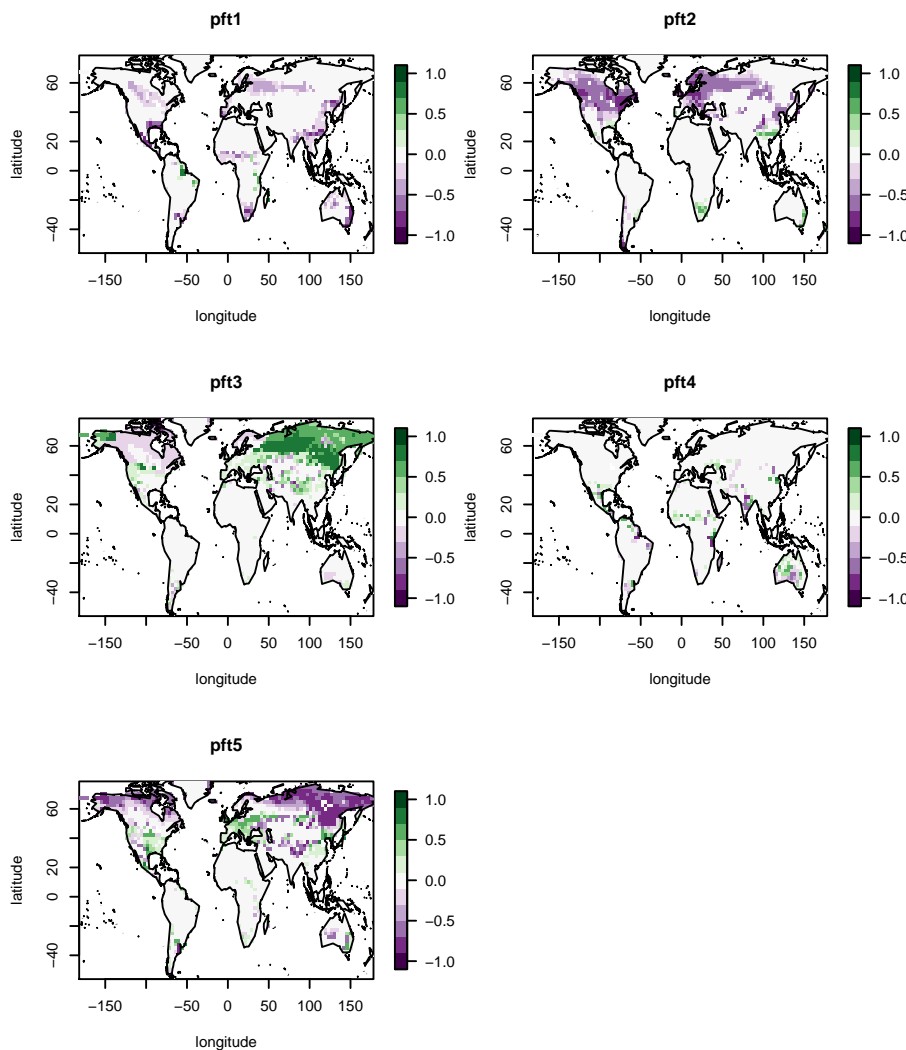

**Figure 10.** Maps of mean annual anomalies of vegetation cover between LGM and PI in the Dynamic simulations, for the five PFTs. PFT1 is broadleaf trees; PFT2 is needleleaf trees; PFT3 is C3 grasses; PFT4 is C4 grasses; PFT5 is shrubs.

Bigelow, N. H., Brubaker, L. B., Edwards, M. E., Harrison, S. P., Prentice, I. C., Anderson, P. M., Andreev, A. A., Bartlein, P. J., Christensen, T. R., Cramer, W., Kaplan, J. O., Lozhkin, A. V., Matveyeva, N. V., Murray, D. F., McGuire, A. D., Razzhivin, V. Y., Ritchie, J. C., Smith, B., Walker, D. A., Gajewski, K., Wolf, V., Holmqvist, B. H., Igarashi, Y., Kremenetskii, K., Paus, A., Pisaric, M. F. J., and Volkova, V. S.: Climate change and Arctic ecosystems: 1. Vegetation changes north of 55°N between the last glacial maximum, mid-Holocene, and present, Journal of Geophysical Research: Atmospheres, 108, n/a–n/a, doi:10.1029/2002JD002558, http://dx.doi.org/10.1029/2002JD002558, 8170, 2003.

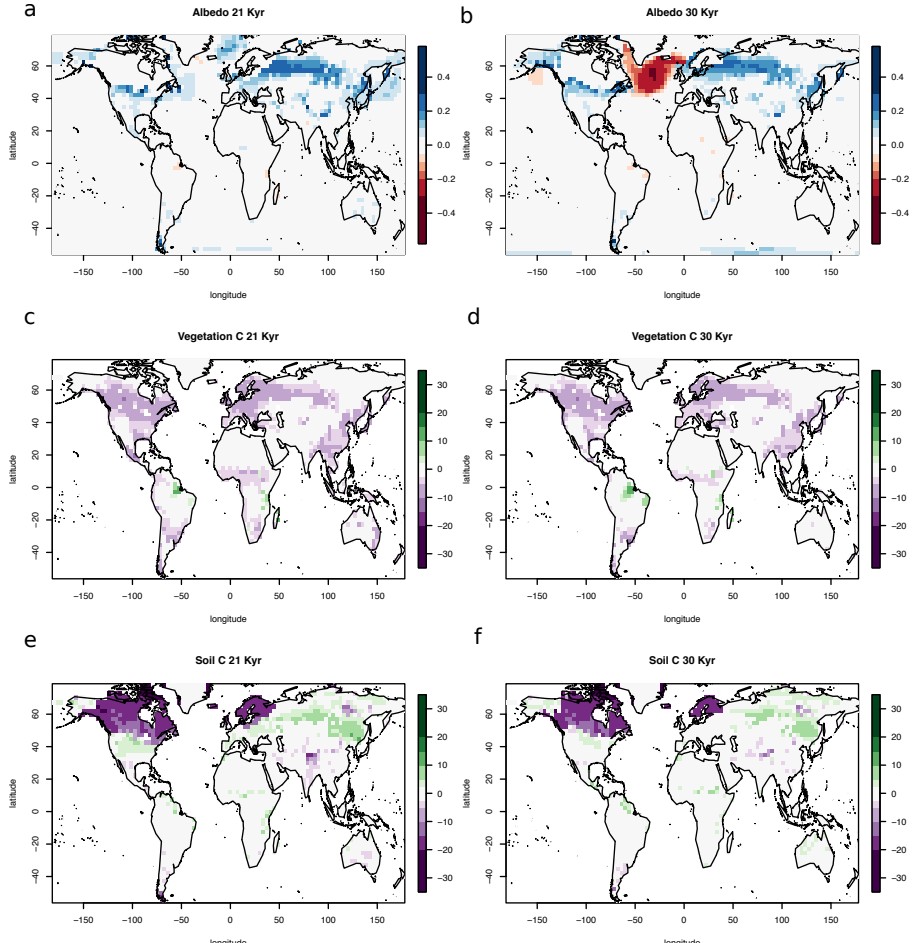

**Figure 11.** Mean annual anomaly of: a) and b), surface albedo (unitless); c) and d), vegetation carbon (kg C m$^{-2}$); and e) and f), soil carbon (kg C m$^{-2}$), between the Dynamic Vegetation simulation and the equivalent Static Vegetation simulation. For 21 ka (a, c, and e) and 30 ka (b, d, and f). The pattern of reduced surface albedo at 30 ka is similar to the pattern at 56 ka and 100 ka. Since in the Static simulations the carbon remains at PI levels, figures c - f also represent the anomaly to PI. This figure includes the carbon under ice-sheets.

Bird, M. I., Lloyd, J., and Farquhar, G. D.: Terrestrial carbon storage at the LGM, Nature, 371, 566–566, doi:10.1038/371566a0, http://www.nature.com/nature/journal/v371/n6498/abs/371566a0.html, 1994.

Bonan, G. B.: Forests and Climate Change: Forcings, Feedbacks, and the Climate Benefits of Forests, Science, 320, 1444–1449, doi:10.1126/science.1155121, http://www.sciencemag.org/content/320/5882/1444, 2008.

5   Bradshaw, C. D., Lunt, D. J., Flecker, R., and Davies-Barnard, T.: Disentangling the roles of late Miocene palaeogeography and vegetation Implications for climate sensitivity, Palaeogeography, Palaeoclimatology, Palaeoecology, 417, 17–34, doi:10.1016/j.palaeo.2014.10.003, http://www.sciencedirect.com/science/article/pii/S0031018214004908, 2015.

Brovkin, V., Ganopolski, A., Archer, D., and Munhoven, G.: Glacial CO2 cycle as a succession of key physical and biogeochemical processes, Clim. Past, 8, 251–264, doi:10.5194/cp-8-251-2012, http://www.clim-past.net/8/251/2012/, 2012.

Brovkin, V., Boysen, L., Arora, V. K., Boisier, J. P., Cadule, P., Chini, L., Claussen, M., Friedlingstein, P., Gayler, V., van den Hurk, B. J. J. M., Hurtt, G. C., Jones, C. D., Kato, E., de Noblet-Ducoudré, N., Pacifico, F., Pongratz, J., and Weiss, M.: Effect of Anthropogenic Land-Use and Land-Cover Changes on Climate and Land Carbon Storage in CMIP5 Projections for the Twenty-First Century, Journal of Climate, 26, 6859–6881, doi:10.1175/JCLI-D-12-00623.1, http://journals.ametsoc.org/doi/abs/10.1175/JCLI-D-12-00623.1, 2013a.

Brovkin, V., Boysen, L., Raddatz, T., Gayler, V., Loew, A., and Claussen, M.: Evaluation of vegetation cover and land-surface albedo in MPI-ESM CMIP5 simulations, Journal of Advances in Modeling Earth Systems, pp. n/a–n/a, doi:10.1029/2012MS000169, http://onlinelibrary.wiley.com/doi/10.1029/2012MS000169/abstract, 2013b.

Ciais, P., Tagliabue, A., Cuntz, M., Bopp, L., Scholze, M., Hoffmann, G., Lourantou, A., Harrison, S. P., Prentice, I. C., Kelley, D. I., Koven, C., and Piao, S. L.: Large inert carbon pool in the terrestrial biosphere during the Last Glacial Maximum, Nature Geoscience, 5, 74–79, doi:10.1038/ngeo1324, http://www.nature.com/ngeo/journal/v5/n1/abs/ngeo1324.html, 2012.

Claussen, M.: Late Quaternary vegetation-climate feedbacks, Clim. Past, 5, 203–216, doi:10.5194/cp-5-203-2009, http://www.clim-past.net/5/203/2009/, 2009.

Claussen, M., Fohlmeister, J., Ganopolski, A., and Brovkin, V.: Vegetation dynamics amplifies precessional forcing, Geophysical Research Letters, 33, L09 709, doi:10.1029/2006GL026111, http://onlinelibrary.wiley.com/doi/10.1029/2006GL026111/abstract, 2006.

Colbourn, G., Ridgwell, A., and Lenton, T. M.: The Rock Geochemical Model (RokGeM) v0.9, Geoscientific Model Development, 6, 1543–1573, doi:10.5194/gmd-6-1543-2013, http://www.geosci-model-dev.net/6/1543/2013/, 2013.

Collins, M., Tett, S. F. B., and Cooper, C.: The internal climate variability of HadCM3, a version of the Hadley Centre coupled model without flux adjustments, Climate Dynamics, 17, 61–81, doi:10.1007/s003820000094, http://link.springer.com/article/10.1007/s003820000094, 2001.

Cowling, S. A., Maslin, M. A., and Sykes, M. T.: Paleovegetation Simulations of Lowland Amazonia and Implications for Neotropical Allopatry and Speciation, Quaternary Research, 55, 140 – 149, doi:http://dx.doi.org/10.1006/qres.2000.2197, http://www.sciencedirect.com/science/article/pii/S0033589400921975, 2001.

Cox, P., Huntingford, C., and Harding, R.: A canopy conductance and photosynthesis model for use in a GCM land surface scheme, Journal of Hydrology, 212-213, 79–94, doi:10.1016/S0022-1694(98)00203-0, http://www.sciencedirect.com/science/article/pii/S0022169498002030, 1998.

Cox, P. M.: Description of the TRIFFID dynamic global vegetation model, http://www.metoffice.gov.uk/media/pdf/9/h/HCTN_24.pdf, 2001.

Cox, P. M., Betts, R. A., Bunton, C. B., Essery, R. L. H., Rowntree, P. R., and Smith, J.: The impact of new land surface physics on the GCM simulation of climate and climate sensitivity, Climate Dynamics, 15, 183–203, doi:10.1007/s003820050276, http://www.springerlink.com/content/9b459pyfhyjwk1ln/abstract/, 1999.

Crowley, T. J.: Ice Age terrestrial carbon changes revisited, Global Biogeochemical Cycles, 9, 377–389, doi:10.1029/95GB01107, http://dx.doi.org/10.1029/95GB01107, 1995.

Crucifix, M. and Loutre, F. M.: Transient simulations over the last interglacial period (1260-115 kyr BP): feedback and forcing analysis, Climate Dynamics, 19, 417–433, doi:10.1007/s00382-002-0234-z, http://link.springer.com/article/10.1007/s00382-002-0234-z, 2002.

Davies-Barnard, T., Valdes, P. J., Singarayer, J. S., and Jones, C. D.: Climatic impacts of land-use change due to crop yield increases and a universal carbon tax from a scenario model, Journal of Climate, 27, 1413–1424, doi:10.1175/JCLI-D-13-00154.1, http://journals.ametsoc.org/doi/abs/10.1175/JCLI-D-13-00154.1, 2014a.

Davies-Barnard, T., Valdes, P. J., Singarayer, J. S., Pacifico, F. M., and Jones, C. D.: Full effects of land use change in the representative concentration pathways, Environmental Research Letters, 9, 114 014, doi:10.1088/1748-9326/9/11/114014, http://iopscience.iop.org/1748-9326/9/11/114014, 2014b.

Davies-Barnard, T., Valdes, P. J., Singarayer, J. S., Wiltshire, A. J., and Jones, C. D.: Quantifying the relative importance of land cover change from climate and land use in the representative concentration pathways, Global Biogeochemical Cycles, p. 2014GB004949, doi:10.1002/2014GB004949, http://onlinelibrary.wiley.com/doi/10.1002/2014GB004949/abstract, 2015.

Davin, E. L. and de Noblet-Ducoudré, N.: Climatic Impact of Global-Scale Deforestation: Radiative versus Nonradiative Processes, Journal of Climate, 23, 97–112, doi:10.1175/2009JCLI3102.1, http://journals.ametsoc.org/doi/abs/10.1175/2009JCLI3102.1, 2010.

de Noblet, N. I., Prentice, I. C., Joussaume, S., Texier, D., Botta, A., and Haxeltine, A.: Possible role of atmosphere-biosphere interactions in triggering the Last Glaciation, Geophysical Research Letters, 23, 3191–3194, doi:10.1029/96GL03004, http://onlinelibrary.wiley.com/doi/10.1029/96GL03004/abstract, 1996.

Edwards, N. R. and Marsh, R.: Uncertainties due to transport-parameter sensitivity in an efficient 3-D ocean-climate model, Climate Dynamics, 24, 415–433, doi:10.1007/s00382-004-0508-8, http://dx.doi.org/10.1007/s00382-004-0508-8, 2005.

Essery, R., Best, M., and Cox, P.: MOSES 2.2 technical documentation, http://biodav.atmos.colostate.edu/kraus/Papers/Biosphere%20Models/HCTN_30.pdf, 2001.

Gallimore, R. G. and Kutzbach, J. E.: Role of orbitally induced changes in tundra area in the onset of glaciation, Nature, 381, 503–505, http://dx.doi.org/10.1038/381503a0, 1996.

Gasse, F.: Hydrological changes in the African tropics since the Last Glacial Maximum, Quaternary Science Reviews, 19, 189 – 211, doi:https://doi.org/10.1016/S0277-3791(99)00061-X, http://www.sciencedirect.com/science/article/pii/S027737919900061X, 2000.

Gedney, N. and Valdes, P. J.: The effect of Amazonian deforestation on the northern hemisphere circulation and climate, Geophysical Research Letters, 27, 3053–3056, doi:10.1029/2000GL011794, http://onlinelibrary.wiley.com/doi/10.1029/2000GL011794/abstract, 2000.

Gillett, N. P., Arora, V. K., Matthews, D., and Allen, M. R.: Constraining the ratio of global warming to cumulative $CO_2$ emissions using CMIP5 simulations, Journal of Climate, p. 130314153438000, doi:10.1175/JCLI-D-12-00476.1, http://journals.ametsoc.org/doi/abs/10.1175/JCLI-D-12-00476.1, 2013.

Goodwin, P., Williams, R. G., and Ridgwell, A.: Sensitivity of climate to cumulative carbon emissions due to compensation of ocean heat and carbon uptake, Nature Geoscience, 8, 29–34, doi:10.1038/ngeo2304, http://www.nature.com/ngeo/journal/v8/n1/abs/ngeo2304.html, 2015.

Gregory, D., Smith, R. N. B., and Cox, P. M.: CANOPY, SURFACE AND SOIL HYDROLOGY, http://precis.metoffice.com/UM_Docs/025.pdf, 1994.

Harrison, S. P. and Prentice, C. I.: Climate and CO2 controls on global vegetation distribution at the last glacial maximum: analysis based on palaeovegetation data, biome modelling and palaeoclimate simulations, Global Change Biology, 9, 983–1004, doi:10.1046/j.1365-2486.2003.00640.x, http://onlinelibrary.wiley.com/doi/10.1046/j.1365-2486.2003.00640.x/abstract, 2003.

Harrison, S. P., Yu, G., Takahara, H., and Prentice, I. C.: Palaeovegetation (Communications arising): Diversity of temperate plants in east Asia, Nature, 413, 129–130, http://dx.doi.org/10.1038/35093166, 2001.

Hoogakker, B. A. A., Smith, R. S., Singarayer, J. S., Marchant, R., Prentice, I. C., Allen, J. R. M., Anderson, R. S., Bhagwat, S. A., Behling, H., Borisova, O., Bush, M., Correa-Metrio, A., de Vernal, A., Finch, J. M., Fréchette, B., Lozano-Garcia, S., Gosling, W. D., Granoszewski, W., Grimm, E. C., Grüger, E., Hanselman, J., Harrison, S. P., Hill, T. R., Huntley, B., Jiménez-Moreno, G., Kershaw, P., Ledru, M.-P., Magri, D., McKenzie, M., Müller, U., Nakagawa, T., Novenko, E., Penny, D., Sadori, L., Scott, L., Stevenson, J., Valdes, P. J., Vandergoes,

M., Velichko, A., Whitlock, C., and Tzedakis, C.: Terrestrial biosphere changes over the last 120 kyr, Climate of the Past, 12, 51–73, doi:10.5194/cp-12-51-2016, http://www.clim-past.net/12/51/2016/, 2016.

Hopcroft, P. O. and Valdes, P. J.: Last glacial maximum constraints on the Earth System model HadGEM2-ES, Climate Dynamics, 45, 1657–1672, doi:10.1007/s00382-014-2421-0, http://link.springer.com/article/10.1007/s00382-014-2421-0, 2014.

Hugelius, G., Strauss, J., Zubrzycki, S., Harden, J. W., Schuur, E. A. G., Ping, C.-L., Schirrmeister, L., Grosse, G., Michaelson, G. J., Koven, C. D., O'Donnell, J. A., Elberling, B., Mishra, U., Camill, P., Yu, Z., Palmtag, J., and Kuhry, P.: Estimated stocks of circumpolar permafrost carbon with quantified uncertainty ranges and identified data gaps, Biogeosciences, 11, 6573–6593, doi:10.5194/bg-11-6573-2014, http://www.biogeosciences.net/11/6573/2014/, 2014.

Jahn, A., Claussen, M., Ganopolski, A., and Brovkin, V.: Quantifying the effect of vegetation dynamics on the climate of the Last Glacial
Maximum, Clim. Past, 1, 1–7, doi:10.5194/cp-1-1-2005, http://www.clim-past.net/1/1/2005/, 2005.

Kageyama, M., Braconnot, P., Bopp, L., Caubel, A., Foujols, M.-A., Guilyardi, E., Khodri, M., Lloyd, J., Lombard, F., Mariotti, V., Marti, O., Roy, T., and Woillez, M.-N.: Mid-Holocene and Last Glacial Maximum climate simulations with the IPSL model-part I: comparing IPSL_CM5A to IPSL_CM4, Climate Dynamics, 40, 2447–2468, doi:10.1007/s00382-012-1488-8, http://link.springer.com/article/10.1007/s00382-012-1488-8, 2012.

Kaplan, J. O.: Wetlands at the Last Glacial Maximum: Distribution and methane emissions, Geophysical Research Letters, 29, 3–1–3–4, doi:10.1029/2001GL013366, http://dx.doi.org/10.1029/2001GL013366, 2002.

Kaplan, J. O., Prentice, I. C., Knorr, W., and Valdes, P. J.: Modeling the dynamics of terrestrial carbon storage since the Last Glacial Maximum, Geophysical Research Letters, 29, 2074, doi:10.1029/2002GL015230, http://onlinelibrary.wiley.com/doi/10.1029/2002GL015230/abstract, 2002.

Kohfeld, K. E. and Ridgwell, A.: Glacial-Interglacial Variability in Atmospheric CO2, in: Surface Ocean Lower Atmosphere Processes, edited by Quéré, C. L. and Saltzman, E. S., pp. 251–286, American Geophysical Union, http://onlinelibrary.wiley.com/doi/10.1029/2008GM000845/summary, 2009.

Köhler, P. and Fischer, H.: Simulating changes in the terrestrial biosphere during the last glacial/interglacial transition, Global and Planetary Change, 43, 33 – 55, doi:http://dx.doi.org/10.1016/j.gloplacha.2004.02.005, http://www.sciencedirect.com/science/article/pii/
S0921818104000542, 2004.

Köhler, P., Knorr, G., and Bard, E.: Permafrost thawing as a possible source of abrupt carbon release at the onset of the Bolling/Allerod, Nature Communications, 5, 5520, doi:10.1038/ncomms6520, http://www.nature.com/ncomms/2014/141120/ncomms6520/full/ncomms6520.html, 2014.

Lawson, E. C., Wadham, J. L., Tranter, M., Stibal, M., Lis, G. P., Butler, C. E. H., Laybourn-Parry, J., Nienow, P., Chandler, D., and Dewsbury,
P.: Greenland Ice Sheet exports labile organic carbon to the Arctic oceans, Biogeosciences, 11, 4015–4028, doi:10.5194/bg-11-4015-2014, http://www.biogeosciences.net/11/4015/2014/, 2014.

Lord, N. S., Ridgwell, A., Thorne, M. C., and Lunt, D. J.: An impulse response function for the long tail of excess atmospheric CO2 in an Earth system model, Global Biogeochemical Cycles, 30, 2–17, doi:10.1002/2014GB005074, http://dx.doi.org/10.1002/2014GB005074, 2014GB005074, 2016.

Martin Calvo, M. and Prentice, I. C.: Effects of fire and CO2 on biogeography and primary production in glacial and modern climates, New Phytologist, 208, 987–994, doi:10.1111/nph.13485, http://dx.doi.org/10.1111/nph.13485, 2014-18722, 2015.

Maslin, M. A., Ettwein, V. J., Boot, C. S., Bendle, J., and Pancost, R. D.: Amazon Fan biomarker evidence against the Pleistocene rainforest refuge hypothesis?, Journal of Quaternary Science, 27, 451–460, doi:10.1002/jqs.1567, http://dx.doi.org/10.1002/jqs.1567, 2012.

Matthews, H. D., Weaver, A. J., Eby, M., and Meissner, K. J.: Radiative forcing of climate by historical land cover change, Geophysical Research Letters, 30, 1055, doi:10.1029/2002GL016098, http://onlinelibrary.wiley.com/doi/10.1029/2002GL016098/abstract, 2003.

Matthews, H. D., Gillett, N. P., Stott, P. A., and Zickfeld, K.: The proportionality of global warming to cumulative carbon emissions, Nature, 459, 829–832, doi:10.1038/nature08047, http://www.nature.com/nature/journal/v459/n7248/full/nature08047.html, 2009.

Menviel, L., Yu, J., Joos, F., Mouchet, A., Meissner, K. J., and England, M. H.: Poorly ventilated deep ocean at the Last Glacial Maximum inferred from carbon isotopes: A data-model comparison study, Paleoceanography, 32, 2–17, doi:10.1002/2016PA003024, http://dx.doi.org/10.1002/2016PA003024, 2016PA003024, 2017.

Montenegro, A., Eby, M., Kaplan, J. O., Meissner, K. J., and Weaver, A. J.: Carbon storage on exposed continental shelves during the glacial-interglacial transition, Geophysical Research Letters, 33, n/a–n/a, doi:10.1029/2005GL025480, http://dx.doi.org/10.1029/2005GL025480,
l08703, 2006.

Niessen, F., Hong, J. K., Hegewald, A., Matthiessen, J., Stein, R., Kim, H., Kim, S., Jensen, L., Jokat, W., Nam, S.-I., and Kang, S.-H.: Repeated Pleistocene glaciation of the East Siberian continental margin, Nature Geoscience, 6, 842–846, doi:10.1038/ngeo1904, http://www.nature.com/ngeo/journal/v6/n10/abs/ngeo1904.html, 2013.

O'ishi, R. and Abe-Ouchi, A.: Influence of dynamic vegetation on climate change and terrestrial carbon storage in the Last Glacial Maximum,
Clim. Past, 9, 1571–1587, doi:10.5194/cp-9-1571-2013, http://www.clim-past.net/9/1571/2013/, 2013.

Pickett, E. J., Harrison, S. P., Hope, G., Harle, K., Dodson, J. R., Peter Kershaw, A., Colin Prentice, I., Backhouse, J., Colhoun, E. A., D'Costa, D., Flenley, J., Grindrod, J., Haberle, S., Hassell, C., Kenyon, C., Macphail, M., Martin, H., Martin, A. H., McKenzie, M., Newsome, J. C., Penny, D., Powell, J., Ian Raine, J., Southern, W., Stevenson, J., Sutra, J.-P., Thomas, I., van der Kaars, S., and Ward, J.: Pollen-based reconstructions of biome distributions for Australia, Southeast Asia and the Pacific (SEAPAC region) at 0, 6000 and 18,000 14C yr BP,
Journal of Biogeography, 31, 1381–1444, doi:10.1111/j.1365-2699.2004.01001.x, http://dx.doi.org/10.1111/j.1365-2699.2004.01001.x, 2004.

Pongratz, J., Reick, C. H., Raddatz, T., and Claussen, M.: Biogeophysical versus biogeochemical climate response to historical anthropogenic land cover change, Geophysical Research Letters, 37, L08 702, doi:10.1029/2010GL043010, http://onlinelibrary.wiley.com/doi/10.1029/2010GL043010/abstract, 2010.

Pope, V. D., Gallani, M. L., Rowntree, P. R., and Stratton, R. A.: The impact of new physical parametrizations in the Hadley Centre climate model: HadAM3, Climate Dynamics, 16, 123–146, http://dx.doi.org/10.1007/s003820050009, 2000.

Prentice, I. C. and Jolly, D.: Mid-Holocene and glacial-maximum vegetation geography of the northern continents and Africa, Journal of Biogeography, 27, 507–519, doi:10.1046/j.1365-2699.2000.00425.x, http://dx.doi.org/10.1046/j.1365-2699.2000.00425.x, 2000.

Prentice, I. C., Sykes, M., Lautenschlager, M., Harrison, S., Denissenko, O., and Bartlein, P.: Modelling Global Vegetation Patterns and
Terrestrial Carbon Storage at the Last Glacial Maximum, Global Ecology and Biogeography Letters, 3, 67–76, http://www.jstor.org/stable/2997548, 1993.

Prentice, I. C., Harrison, S. P., and Bartlein, P. J.: Global vegetation and terrestrial carbon cycle changes after the last ice age, New Phytologist, 189, 988–998, doi:10.1111/j.1469-8137.2010.03620.x, http://dx.doi.org/10.1111/j.1469-8137.2010.03620.x, 2011.

Ridgwell, A. and Hargreaves, J. C.: Regulation of atmospheric CO2 by deep-sea sediments in an Earth system model, Global Biogeochemical
Cycles, 21, n/a–n/a, doi:10.1029/2006GB002764, http://dx.doi.org/10.1029/2006GB002764, gB2008, 2007.

Ridgwell, A., Hargreaves, J. C., Edwards, N. R., Annan, J. D., Lenton, T. M., Marsh, R., Yool, A., and Watson, A.: Marine geochemical data assimilation in an efficient Earth System Model of global biogeochemical cycling, Biogeosciences, 4, 87–104, doi:10.5194/bg-4-87-2007, http://www.biogeosciences.net/4/87/2007/, 2007.

Ridgwell, A., Maslin, M., and Kaplan, J. O.: Flooding of the continental shelves as a contributor to deglacial CH4 rise, Journal of Quaternary Science, 27, 800–806, doi:10.1002/jqs.2568, http://onlinelibrary.wiley.com/doi/10.1002/jqs.2568/abstract, 2012.

Saito, K., Sueyoshi, T., Marchenko, S., Romanovsky, V., Otto-Bliesner, B., Walsh, J., Bigelow, N., Hendricks, A., and Yoshikawa, K.: LGM permafrost distribution: how well can the latest PMIP multi-model ensembles perform reconstruction?, Climate of the Past, 9, 1697–1714, doi:10.5194/cp-9-1697-2013, http://www.clim-past.net/9/1697/2013/, 2013.

Shackleton, N. J., Lamb, H. H., Worssam, B. C., Hodgson, J. M., Lord, A. R., Shotton, F. W., Schove, D. J., and Cooper, L. H. N.: The Oxygen Isotope Stratigraphic Record of the Late Pleistocene [and Discussion], Philosophical Transactions of the Royal Society of London B: Biological Sciences, 280, 169–182, doi:10.1098/rstb.1977.0104, http://rstb.royalsocietypublishing.org/content/280/972/169, 1977.

Shellito, C. J. and Sloan, L. C.: Reconstructing a lost Eocene Paradise, Part II: On the utility of dynamic global vegetation models in pre-Quaternary climate studies, Global and Planetary Change, 50, 18 – 32, doi:http://dx.doi.org/10.1016/j.gloplacha.2005.08.002, http://www.sciencedirect.com/science/article/pii/S0921818105001487, 2006.

Singarayer, J. S. and Valdes, P. J.: High-latitude climate sensitivity to ice-sheet forcing over the last 120 kyr, Quaternary Science Reviews, 29, 43 – 55, doi:http://dx.doi.org/10.1016/j.quascirev.2009.10.011, http://www.sciencedirect.com/science/article/pii/S0277379109003564, climate of the Last Million Years: New Insights from {EPICA} and Other Records, 2010.

Singarayer, J. S., Ridgwell, A., and Irvine, P.: Assessing the benefits of crop albedo bio-geoengineering, Environmental Research Letters, 4, 045 110, http://stacks.iop.org/1748-9326/4/i=4/a=045110, 2009.

Singarayer, J. S., Valdes, P. J., Friedlingstein, P., Nelson, S., and Beerling, D. J.: Late Holocene methane rise caused by orbitally controlled increase in tropical sources, Nature, 470, 723–757, doi:10.1038/nature09739, http://www.nature.com/nature/journal/v470/n7332/abs/nature09739.html, 2011.

Valdes, P. J., Armstrong, E., Badger, M. P. S., Bradshaw, C. D., Bragg, F., Davies-Barnard, T., Day, J. J., Farnsworth, A., Hopcroft, P. O., Kennedy, A. T., Lord, N. S., Lunt, D. J., Marzocchi, A., Parry, L. M., Roberts, W. H. G., Stone, E. J., Tourte, G. J. L., and Williams, J. H. T.: The BRIDGE HadCM3 family of climate models: HadCM3@Bristol v1.0, Geosci. Model Dev. Discuss., 2017, 1–42, doi:10.5194/gmd-2017-16, http://www.geosci-model-dev-discuss.net/gmd-2017-16/, 2017.

Wadham, J. L., Arndt, S., Tulaczyk, S., Stibal, M., Tranter, M., Telling, J., Lis, G. P., Lawson, E., Ridgwell, A., Dubnick, A., Sharp, M. J., Anesio, A. M., and Butler, C. E. H.: Potential methane reservoirs beneath Antarctica, Nature, 488, 633–637, doi:10.1038/nature11374, http://www.nature.com/nature/journal/v488/n7413/abs/nature11374.html, 2012.

Wang, X., Edwards, R. L., Auler, A. S., Cheng, H., Kong, X., Wang, Y., Cruz, F. W., Dorale, J. A., and Chiang, H.-W.: Hydroclimate changes across the Amazon lowlands over the past 45,000 years, Nature, 541, 204–207, doi:10.1038/nature20787, http://www.nature.com/doifinder/10.1038/nature20787, 2017.

Zeebe, R. E., Wolf-Gladrow, D., and Jansen, H.: On the time required to establish chemical and isotopic equilibrium in the carbon dioxide system in seawater, Marine Chemistry, 65, 135 – 153, doi:https://doi.org/10.1016/S0304-4203(98)00092-9, http://www.sciencedirect.com/science/article/pii/S0304420398000929, 1999.

Zeng, N.: Glacial-interglacial atmospheric CO2 change - The glacial burial hypothesis, Advances in Atmospheric Sciences, 20, 677–693, doi:10.1007/BF02915395, http://link.springer.com/article/10.1007/BF02915395, 2003.

Zhou, J., Poulsen, C. J., Rosenbloom, N., Shields, C., and Briegleb, B.: Vegetation-climate interactions in the warm mid-Cretaceous, Clim. Past, 8, 565–576, doi:10.5194/cp-8-565-2012, http://www.clim-past.net/8/565/2012/, 2012.

Zimov, N. S., Zimov, S. A., Zimova, A. E., Zimova, G. M., Chuprynin, V. I., and Chapin, F. S.: Carbon storage in permafrost and soils of the mammoth tundra-steppe biome: Role in the global carbon budget, Geophysical Research Letters, 36, L02 502, doi:10.1029/2008GL036332, http://dx.doi.org/10.1029/2008GL036332, l02502, 2009.

Zimov, S. A., Schuur, E. A. G., and Chapin, F. S.: Permafrost and the Global Carbon Budget, Science, 312, 1612–1613, doi:10.1126/science.1128908, http://science.sciencemag.org/content/312/5780/1612, 2006.

