# Peer review of "Quantifying the Influence of the Terrestrial Biosphere on Glacial-interglacial Climate Dynamics"

_Climate of the Past, 2017_

## Short Comment (SC1) · 17 Feb 2017

**Interesting simulations, but what about the real world?**

The Abstract severely overestimates the uncertainty in estimates of the change in land carbon storage between the last glacial maximum (LGM) and pre-industrial time. The impression is given that the biogeochemical consequences of land-ocean transfer are small, and even of unknown sign. Hence the last sentence of the Abstract, which implies (unjustifiably in my view) that land biogeochemical feedbacks can be neglected on these time scales.

The problem arises as soon as the range of modelled values for the LGM − PI difference in land carbon storage is given as −440 to +37 PgC. In fact, only the value of −440 PgC is defensible. The other three values calculated assume either that there was no vegetation on the exposed continental shelves, and/or that pre-existing vegetation and soil carbon was not transferred to the ocean but somehow remained *in situ*. The main text makes clear that the authors do not, in fact, consider these as likely alternatives; so they should not be given equal weight.

The problem is compounded by a superficial treatment of the literature on observationally based estimates of this difference. The current text gives equal credibility to attempts made two decades ago to estimate carbon storage "bottom-up", either via manual interpolation of sparse pollen records (e.g. Adams & Faure 1998, Crowley 1995) or based on climate and biome model simulations (e.g. Prentice et al. 1993) A review of this topic (Prentice & Harrison 2009) noted the unreliability of all of these methods, which (a) assume a constant carbon density per biome and (b) disregard the effect of $CO_2$ concentration on plant productivity and, therefore, carbon storage.

Ciais et al. (2012 – nb, this should be 2011) is cited for the large range of previously published values that is summarized there. But Ciais et al. more importantly provided the most comprehensive analysis of benthic $\delta^{13}C$ data to date, and a complete isotopic mass balance calculation, arriving at a best estimate of −330 PgC. Ciais et al. also attributed the discrepancy between this observationally based estimate and several larger, model-based estimates to the counterbalancing effect of a large inert carbon pool (putatively stored in permafrost) at the LGM. This idea is further supported by the recent work of Crichton et al. (2016) showing that the $\delta^{13}C$ record of atmospheric $CO_2$ over the deglaciation can be well explained by a substantial permafrost carbon contribution to the deglacial $CO_2$ rise.

The authors give too little information about the land biosphere model that they used. In particular, no information is given about the formulation of the $CO_2$ effect on primary production. This effect is critical given the large variations in atmospheric $CO_2$ that occurred during the period studied. The plant functional type maps show unrealistically extensive tropical forests at the LGM (in comparison to pollen data, see e.g. Prentice et al. 2011; and offshore *n*-alkane $\delta^{13}C$ measurements, see e.g. Bragg et al. 2013). This suggests that the model underestimates the effect of low $CO_2$ on global biome distribution and also may mean that the strength of the biogeophysical feedback – a key point of the paper – has been underestimated.

A minor point concerns the East Siberian ice sheet. According to the cited reference, and most other recent treatments, there was no such ice sheet during the last glacial period.

Finally, although the paper makes much of the limited contribution of changes in terrestrial carbon storage to the long term course of atmospheric $CO_2$ concentration in part due to compensating oceanic mechanisms, this is not a new finding. For example, the analysis by Joos et al. (2004) – of which Paul Valdes was a co-author – accounts for the $CaCO_3$ compensation mechanism and indicates a more than six-fold "dampening" of the effect of terrestrial carbon storage changes on atmospheric $CO_2$ over multimillennial time scales.

New references:

Bragg, F., I.C. Prentice, S.P. Harrison, G. Eglinton, P.N. Foster, F. Rommerskirchen and J. Rullkötter (2013) Stable isotope and modelling evidence for $CO_2$ as a driver of glacial-interglacial vegetation shifts in southern Africa. *Biogeosciences* **10**: 2001-2010.

Crichton, K.A., N. Bouttes, D.M. Roche, J. Chapellaz and G. Krinner (2016) Permafrost carbon as a missing link to explain $CO_2$ changes during the last deglaciation. *Nature Geoscience* **9**: 683-686.

Joos, F., S. Gerber, I.C. Prentice, B. L. Otto-Bliesner and P.J. Valdes (2004) Transient simulations of Holocene atmospheric carbon dioxide and terrestrial carbon since the Last Glacial Maximum. *Global Biogeochemical Cycles* **18**: GB2002.

Prentice, I.C. and S.P. Harrison (2009) Ecosystem effects of $CO_2$ concentration: evidence from past climates. *Climate of the Past* **5**: 297-307.

Prentice, I.C., S.P. Harrison and P.J. Bartlein (2011) Global vegetation and terrestrial carbon cycle changes after the last ice age. *New Phytologist* **189**: 988-998.

---

## Author Comment (AC1) · 9 Mar 2017

Response to comment by Colin Prentice: *Interesting simulations, but what about the real world?*

Thank you to Colin for his detailed comments on the LGM carbon aspect of the paper. We note that none of his comments detract from the main point: the balance of biogeophysical and biogeochemical climate effects from the terrestrial biosphere over the glacial-interglacial cycle. After the formal reviews, we will be happy to make some adjustments to the text.

For clarity, we include the comments in italics and our response in standard text below.

*The Abstract severely overestimates the uncertainty in estimates of the change in land*

*carbon storage between the last glacial maximum (LGM) and pre-industrial time. The impression is given that the biogeochemical consequences of land-ocean transfer are small, and even of unknown sign. Hence the last sentence of the Abstract, which implies (unjustifiably in my view) that land biogeochemical feedbacks can be neglected on these time scales.*

An abstract is always a delicate balance between giving the most amount of information possible, fairly representing the results, providing appropriate context, and showing the results most likely to interest potential readers. We freely acknowledge we may not have struck the correct balance. We will be guided by the reviewers on whether it would be better to amend the last sentence of the abstract. However, given this is a research paper rather than a review, we feel it is justified to report the novel results and their possible implications, rather than reiterating existing understanding.

*The problem arises as soon as the range of modelled values for the LGM – PI difference in land carbon storage is given as –440 to +37 PgC. In fact, only the value of –440 PgC is defensible. The other three values calculated assume either that there was no vegetation on the exposed continental shelves, and/or that pre-existing vegetation and soil carbon was not transferred to the ocean but somehow remained in situ.*

The two 'no carbon on expanded land area' scenarios assume that soil and vegetation carbon on exposed continental shelves is not returned to the atmosphere when the land is inundated, not that it doesn't exist at all or that it isn't transferred to the ocean. The same is true for the ice-sheets. We agree that this is not made sufficiently clear and will clarify the text accordingly.

*The main text makes clear that the authors do not, in fact, consider these as likely alternatives; so they should not be given equal weight.*

We agree the most likely scenario should be given more weight, and consequently we use the -440 value for further calculations throughout the paper. As we discuss in the paper, the fate of carbon under ice-sheets and after inundation is currently not fully

known. Therefore, we feel it would be disingenuous to dismiss this range all together. Ultimately, our responsibility as researchers is to present the results we found, including the uncertainties, and explain them as best we are able.

*The problem is compounded by a superficial treatment of the literature on observationally based estimates of this difference. The current text gives equal credibility to attempts made two decades ago to estimate carbon storage "bottomup", either via manual interpolation of sparse pollen records (e.g. Adams Faure 1998, Crowley 1995) or based on climate and biome model simulations (e.g. Prentice et al. 1993) A review of this topic (Prentice Harrison 2009) noted the unreliability of all of these methods, which (a) assume a constant carbon density per biome and (b) disregard the effect of CO2 concentration on plant productivity and, therefore, carbon storage*

While we agree that a comprehensive assessment of the biogeochemical, biogeophysical, and vegetation literature for the whole of the 120 ka period would be ideal, space constraints mean we have picked a representative selection. In the introduction we provide context in the form of older papers, and in the discussion we cite a variety of newer papers as comparators. Our intention is to highlight areas of similarity and difference to other research across the several topical and temporal areas we cover, rather than provide a detailed review of any one aspect.

*Ciais et al. (2012 – nb, this should be 2011)*

We believe this misunderstanding originates from the fact that the paper was published online November 2011, whereas the journal itself was published in January 2012. Following the citation provided by Nature Geoscience, we use 2012.

*is cited for the large range of previously published values that is summarized there. But Ciais et al. more importantly provided the most comprehensive analysis of benthic $\delta 13C$ data to date, and a complete isotopic mass balance calculation, arriving at a best estimate of –330 PgC. Ciais et al. also attributed the discrepancy between this observationally based estimate and several larger, model-based estimates to the*

*counterbalancing effect of a large inert carbon pool (putatively stored in permafrost) at the LGM. This idea is further supported by the recent work of Crichton et al. (2016) showing that the δ13C record of atmospheric CO2 over the deglaciation can be well explained by a substantial permafrost carbon contribution to the deglacial CO2 rise.*

We primarily use the -440 PgC value, which is fairly consistent with Ciais et al. (2012) and Crichton et al. (2016). A best estimate is, however, not 'the truth' and does not invalidate exploration of the source and scale of uncertainties. We feel it is important to present the results as we found them, and that it would be wrong to exclude results which do not exactly conform to current thinking.

*The authors give too little information about the land biosphere model that they used. In particular, no information is given about the formulation of the CO2 effect on primary production.*

The dynamic vegetation model used in HadCM3, TRIFFID, is well established, having been first published in Cox et al., (2000), and is the core of the terrestrial carbon cycle in JULES (Clark et al., 2011). TRIFFID has been used in many publications (e.g. Armstrong et al., 2016; Booth et al., 2012; Cox et al., 2004, 2000; Davies-Barnard et al., 2014; Falloon et al., 2012; Friedlingstein et al., 2006; Good et al., 2012; McCarthy et al., 2012). In this paper, we provide five references for the land surface scheme; a summary of the key features of TRIFFID; and a separate reference for the model configurations we use. We feel this is sufficient guidance for interested readers to read further, whilst not overburdening the paper with detail.

*This effect is critical given the large variations in atmospheric CO2 that occurred during the period studied. The plant functional type maps show unrealistically extensive tropical forests at the LGM (in comparison to pollen data, see e.g. Prentice et al. 2011; and offshore n-alkane δ13C measurements, see e.g. Bragg et al. 2013). This suggests that the model underestimates the effect of low CO2 on global biome distribution and also may mean that the strength of the biogeophysical feedback – a key point of the*

*paper – has been underestimated.*

You're right when you say that the extent of tropical forest does not perfectly fit the observations. Our main concern, however, is whether the error is of an order of magnitude that would affect our overall findings. The biogeophysical impact at the LGM in these simulations are consistent with those found by other modelling studies. Were we to assume the biogeophysical effect is underestimated in our model because of underestimating the effect of low CO2, it doesn't damage our conclusions, because if the biogeophysical effect is larger, proportionally any biogeochemical effect is smaller. Therefore, although we agree our model is not perfect, we don't think that invalidates our conclusions.

*A minor point concerns the East Siberian ice sheet. According to the cited reference, and most other recent treatments, there was no such ice sheet during the last glacial period.*

As we say in the text, we exclude the East Siberian ice sheet entirely. It is generally agreed that there was no East Siberian ice sheet at the LGM, but there is more uncertainty in the earlier part of the glacial period (see for instance Ehlers et al., (2011). We agree our phrasing doesn't clarify that we refer to uncertainty in the earlier parts of the glacial period, and we will amend the text accordingly.

*Finally, although the paper makes much of the limited contribution of changes in terrestrial carbon storage to the long term course of atmospheric CO2 concentration in part due to compensating oceanic mechanisms, this is not a new finding. For example, the analysis by Joos et al. (2004) – of which Paul Valdes was a co-author – accounts for the CaCO3 compensation mechanism and indicates a more than six-fold "dampening" of the effect of terrestrial carbon storage changes on atmospheric CO2 over multimillennial time scales.*

The main point of this paper is the assessment of the net (biogeophysical and biogeochemical) terrestrial biosphere contribution to the climate. This does necessitate

covering both the biogeochemical and biogeophysical aspects, and as you point out, of course these have already been done to some extent (usually at lower resolution, as is the case for Joos et al. (2004).). In covering these aspects, we think it's useful to highlight interesting features of our simulations. Inevitably, we will use our new data to make some points similar to those made elsewhere. However, that does not diminish the fact that the key novelty of this paper is assessing both the biogeochemical and biogeophysical perspectives together to understand the total influence of the terrestrial biosphere on the climate over the glacial-interglacial period.

References:

Armstrong, E., Valdes, P., House, J., Singarayer, J., 2016. The Role of CO2 and Dynamic Vegetation on the Impact of Temperate Land-Use Change in the HadCM3 Coupled Climate Model. Earth Interact. 20, 1–20. doi:10.1175/EI-D-15-0036.1

Booth, B.B.B., Jones, C.D., Collins, M., Totterdell, I.J., Cox, P.M., Sitch, S., Huntingford, C., Betts, R.A., Harris, G.R., Lloyd, J., 2012. High sensitivity of future global warming to land carbon cycle processes. Environ. Res. Lett. 7, 24002. doi:10.1088/1748-9326/7/2/024002

Clark, D.B., Mercado, L.M., Sitch, S., Jones, C.D., Gedney, N., Best, M.J., Pryor, M., Rooney, G.G., Essery, R.L.H., Blyth, E., Boucher, O., Harding, R.J., Huntingford, C., Cox, P.M., 2011. The Joint UK Land Environment Simulator (JULES), model description – Part 2: Carbon fluxes and vegetation dynamics. Geosci. Model Dev. 4, 701–722. doi:10.5194/gmd-4-701-2011

Cox, P.M., Betts, R.A., Collins, M., Harris, P.P., Huntingford, C., Jones, C.D., 2004. Amazonian forest dieback under climate-carbon cycle projections for the 21st century. Theor. Appl. Climatol. 78, 137–156.

Cox, P.M., Betts, R.A., Jones, C.D., Spall, S.A., Totterdell, I.J., 2000. Acceleration of global warming due to carbon-cycle feedbacks in a coupled climate model. Nature 408,

184–187. doi:10.1038/35041539

Davies-Barnard, T., Valdes, P.J., Jones, C.D., Singarayer, J.S., 2014. Sensitivity of a coupled climate model to canopy interception capacity. Clim. Dyn. 42, 1715–1732. doi:10.1007/s00382-014-2100-1

Ehlers, J., Ehlers, J., Gibbard, P.L., Hughes, P.D., 2011. Quaternary Glaciations - Extent and Chronology: A Closer Look. Elsevier.

Falloon, P.D., Dankers, R., Betts, R.A., Jones, C.D., Booth, B.B.B., Lambert, F.H., 2012. Role of vegetation change in future climate under the A1B scenario and a climate stabilisation scenario, using the HadCM3C Earth system model. Biogeosciences 9, 4739–4756. doi:10.5194/bg-9-4739-2012

Friedlingstein, P., Cox, P., Betts, R., Bopp, L., Von Bloh, W., Brovkin, V., Cadule, P., Doney, S., Eby, M., Fung, I., 2006. Climate-carbon cycle feedback analysis: Results from the C4MIP model intercomparison. J. Clim. 19, 3337–3353.

Good, P., Jones, C., Lowe, J., Betts, R., Gedney, N., 2012. Comparing Tropical Forest Projections from Two Generations of Hadley Centre Earth System Models, HadGEM2-ES and HadCM3LC. J. Clim. 26, 495–511. doi:10.1175/JCLI-D-11-00366.1

McCarthy, M.P., Sanjay, J., Booth, B.B.B., Krishna Kumar, K., Betts, R.A., 2012. The influence of vegetation on the ITCZ and South Asian monsoon in HadCM3. Earth Syst. Dyn. 3, 87–96. doi:10.5194/esd-3-87-2012

---

## Referee Comment (RC1) · Anonymous Referee #1 · 28 Mar 2017

In this study the biogeochemical and biogeophysical effects of vegetation on the climate system are analysed separately over the time span of the last 120 000 years. Spanning the last ice age inception, last glacial and the latest deglaciation, 62 "snapshot"-type (i.e., under constant forcing) simulations were integrated, distributed over that period, with HadCM3, a fully coupled atmosphere-ocean GCM with interactive vegetation. In addition, 5 transient simulations were integrated over the whole period with various versions of the cGENIE model. These simulations were based on terrestrial carbon fluxes diagnosed from the HadCM3 simulations. The authors conclude that the biogeophysical effects of vegetation account for additional mean cooling during the glacial and, in some cases, substantial regional cooling. The biogeochemical effects are smaller and of opposite sign. The authors also emphasize that different timescales are involved for these two effects on climate.

[Figure]

I have enjoyed reading the paper. It is well written and presents interesting results. I do have three major comments and a long list of other comments and recommend that this paper might be suitable for publication after all of my comments are addressed (i.e. major revisions).

Major comments:

1. One underlying hypothesis with this approach is that the feedbacks considered (ocean, vegetation, atmosphere, sediments, biogeophysical, biogeochemical) add up linearly (i.e. there is no non-linear interaction between the biogeophysical and biogeochemical feedbacks). This assumption needs to be clearly stated and discussed.

2. Under which boundary conditions cGENIE was spun up? Am I right to assume that 120kaBP boundary conditions were used? Am I also right to assume that the diagnosed terrestrial carbon fluxes (calculated based on changes in terrestrial carbon) from the HadCM3 snapshot simulations were interpolated as "emission" forcing time series and applied to cGENIE? And that these "emission" forcing time series were the only changing boundary conditions during the simulations (i.e., no additional imposed $CO_2$ changes, no continental ice sheet changes, etc)? If I understood this right, there is another assumption made by this approach: that the climate sensitivity is constant and independent of the climate state. This assumption also needs to be clearly stated and discussed.

3. As shown in Figure 3b and described on Page 8 (lines 11-15), the vegetation changes trigger a change in ocean circulation. While I agree that this is a model-dependent result and not part of the core results (although certainly influencing these core results), it would be interesting to see time series of AMOC for the static and dynamic simulations. In case there is a simple feedback that can be easily crystallized (such as the one mentioned in the text), it would also be good to analyse the results further, verify that this is indeed the feedback and play and describe this feedback in more detail.

Other comments:

\* Permafrost and wetlands are not (cannot be) resolved correctly. Both play important roles in terrestrial carbon feedbacks. While the Discussion briefly mentions the lack of permafrost related feedbacks in the simulations (Page 14, lines 16-23), it might be interesting to add a discussion about wetlands (changes in wetlands in the tropics, but also changes from permafrost to wetlands or vice-versa at high latitudes). Can you please broaden this discussion, including some key references, and, if possible, add an uncertainty range to your results in the Discussion section.

\* Coarse resolution and numbers of PFTs (page 5, lines 27-32): The representation of vegetation (and therefore associated feedbacks) is crude in TRIFFID (only 5 PFTs, coarse spatial resolution, crude representation of terrestrial nutrient cycles). While this is not any worse than in most other coupled models, the implications should be discussed in more depth in the discussion section.

\* It strikes me in Figure 1 that there is almost no change in tropical rain forest cover. Is that realistic? Would it be possible to include a validation of these results (present day bias + comparison to pollen data from LGM for example)?

\* Table 1: I am certainly misunderstanding something here... Why does the vegetation C change differ for all 4 set-ups? Shouldn't vegetation C only be affected by new land (especially because the atmosphere does not "see" the carbon released from under the ice)? In which case the two ELE simulations should be very similar, if not the same (same for the two ELI simulations)?

\* Page 13, lines 5-8: this is an interesting (although neither surprising nor new) result that feeds back into the discussion about climate sensitivity over long timescales. I would suggest adding a paragraph to the discussion about the different timescales involved and the implications on climate sensitivity. It would also be good to remind the reader, which of these feedbacks are usually incorporated in the state-of-the-art IPCC models (i.e. expand the second paragraph in Discussion).

* HadCM3 simulations: it is my understanding that the HadCM3 simulations are run under constant external forcing, initialized from the previous MOSES 1 simulation, then run for 300 years with equilibrium TRIFFID (50 years of TRIFFID for each 5 years of the climate model run), and finally integrated 300 years with dynamic TRIFFID (where TRIFFID is called every 10 days). I am puzzled by this approach – I would assume that TRIFFID is in equilibrium after the first 300 years of "equilibrium setting". I would also assume that atmospheric conditions are fairly close to equilibrium after these first 300 years, so TRIFFID in "equilibrium setting" saw internal forcing with little drift. Why integrate TRIFFID in dynamic setting for the last 300 years, if the forcing that TRIFFID "sees" is more or less constant in a climatological sense and if TRIFFID is already in equilibrium? This does not change the results presented here, it just seems weird (e.g. why not run TRIFFID in equilibrium over the whole 600 years of constant boundary conditions?). See also text on Page 11, lines 15-16 "the model is run for sufficient length of time for the soil and vegetation carbon to reach equilibrium".

* Set of simulations with static vegetation based on PI simulation (page 4, lines 11-14): I assume that the vegetation in this simulation is masked out under continental ice sheets (adjusted for the ice sheet extension of the period of interest). Can you please add this here.

* Why are the EPICA anomalies halved? Is that to get a representation of global temperature changes (versus changes in local temperature)? Can you please clarify this in the text?

* I do not understand what is shown in Figure 4a. Is it the globally integrated (or globally mean) albedo anomalies? If this is the case; why don't they add up? Or are these means over land/ocean versus global mean? Can you please clarify?

* Figure 5: can you please show these plots for simulations that treat carbon under ice sheets the same way? These plots should only show the difference in C and albedo due to changes in vegetation and soils away from ice sheets. The large purple areas

in 5e and f take the attention away from the results.

* cGENIE simulation TRCE needs to be described in more detail in Section 2.2.

Editorial changes:

* Page 4, line 6: I would prefer to go forward in time: 120 – 0 kaBP. * Page 4, line 21-24: again, please go forward in time; e.g. 120 to 80 kaBP,... * The filled points are hard to see in Figure 2a * Figure 4: I find it hard to discern the different lines – would it be a good idea to plot these time series in different colours? * Page 9, line 6: "is" missing * Figure 6: the red and pink lines are hard to discern * Figure 7: Figure caption should read GCI_ELE and not GLI_ELE

Comment on Colin's comment: I agree with the authors that this is a sensitivity study, showing the whole range of possible scenarios (including unlikely ones). I would not recommend reducing the numbers of scenarios shown and discussed in the text. However, I agree with Colin that it should be made clear in the abstract that the higher terrestrial carbon loss scenarios are more likely (this is the one the cGENIE time series are shown for in Figure 7). Side note: a newer estimate of total terrestrial carbon glacial/interglacial change based on benthic d13C data is given by Menviel et al. (2017).

Menviel, L., Yu, J., Joos, F., Mouchet, A., Meissner, K.J. and M.H. England, 2017: Poorly ventilated deep ocean at the Last Glacial Maximum inferred from carbon isotopes: a data-model comparison study. Paleoceanography, 32, 2-17.

---

## Referee Comment (RC2) · Anonymous Referee #2 · 31 Mar 2017

General comments

In this paper the authors present results of 62 equilibrium simulations with a coupled atmosphere-ocean-vegetation general circulation model covering the last glacial-interglacial cycle. They performed simulations with and without dynamic vegetation to quantify the effect of the terrestrial biosphere on glacial-interglacial climate variability in terms of both biogeophysical and biogeochemical effects. Although the results represent an important contribution to our understanding of the role of the terrestrial biosphere to glacial-interglacial variability, I have some major comments that should be addressed before this paper can be considered for publication in Climate of the Past.

Since changes in vegetation cover are key to the results presented in this study, the modelled vegetation should be compared with available reconstructions, where available. The BIOME6000 dataset for the last glacial maximum provides a unique reconstruction of vegetation cover for glacial climate conditions and could be used to evaluate the model performance. As a minimum requirement, model results and reconstructions should at least be compared qualitatively and discrepancies discussed. Comparison with other modelling studies would also be useful (e.g. (Prentice et al. 2011; Hoogakker et al. 2016)). The representation of vegetation cover in Figure 1 in terms of dominant PFTs can be misleading. For example it seems from Figure 1 that tropical forest remains practically unchanged during glacial conditions. Is this just an artifact of the dominant PFT representation or is it a real feature of the model (in which case the discrepancy with available reconstructions has to be discussed)? In any case I would instead suggest showing fractions of all 5 PFTs separately, at least for pre-industrial and LGM.

The results about the biogeophysical feedback are presented in a rather superficial way, which makes it difficult to get a quantitative understanding of the processes responsible for the positive feedback. As the authors mention in the introduction, the biogeophysical feedback results primarily from the vegetation controlling the surface energy fluxes. The results section in the paper focuses almost exclusively on the effect of changes in surface albedo. Latent heat effects are only mentioned once when referring to Figure 4b but are not discussed further, and sensible heat flux changes are not mentioned at all. Albedo changes are probably the dominant effect, but the other effects should also be quantified. I would suggest to add panels showing the changes in latent and sensible heat flux to Figure 3 and to move the albedo plots from Figure 5 to Figure 3 (it is not clear to me why albedo maps are shown together with vegetation and soil carbon). Also, I would suggest replacing the albedo figures with net shortwave radiation absorbed at the surface. Shortwave radiation is a more appropriate measure because it accounts for changes in insolation and can moreover directly be compared to the latent and sensible heat fluxes.

The authors show that the vegetation is interacting with the thermohaline circulation.

[Figure]

It would be interesting to understand how this is happening. Can anything be said about possible causal relations, given the available simulations? Is vegetation affecting runoff into the North Atlantic, or are vegetation and THC interacting via changes in atmospheric circulation?

In the model description section, no information is given on the soil carbon representation in the model. I expect the results of the biogeochemical part to strongly depend on how soil carbon is represented in the model. In particular, a proper representation of carbon stored in permafrost is probably crucial to model land carbon storage during glacial times. In the discussion section the authors mention that the model does not have a process-based permafrost component, but this should also clearly be stated in the description section. The amount of carbon which can potentially be buried below the ice sheets will strongly depend on how carbon in frozen soils is treated. The authors should discuss this in more detail. How does the carbon stored in permafrost in the model compare to observational estimates (Hugelius et al. 2014) for the present day?

In the biogeochemistry results section the effect of dynamic vegetation on land carbon storage is not discussed, although the differences in vegetation and soil carbon between static and dynamic simulations at LGM is shown in Figure 5 (the 30 kyr maps of vegetation and soil carbon in Figure 5 seem redundant to me). If I understand correctly, in this section only the dynamic vegetation simulations are discussed. This should be clarified in the text.

A figure showing the differences in land carbon storage between LGM and present day would be helpful.

Specific comments

Page 1, line 7: specify that the 62 simulations are 'equilibrium' simulations. Page 1, line 18: ocean/atmosphere Page 3, line 13: vegetation carbon -> land carbon Page 4, line 14: remove brackets Page 4, line 29: timer -> time Figure 1: legend is hard to read

Page 6, line 7: -0.91°C was -0.84°C in the abstract (if I understand correctly) Figure 3: Are the figures for annual mean characteristics? Please specify. Figure 4: use of different colors for different lines would improve readability Page 9, line 6: it IS unclear Page 10, line 2: carbon stores changes -> carbon stores Page 14, lines 8-11: shelf carbon stocks values should be positive Page 14, lines 13-15: check sentence

---

## Author Response (AR1)

**Reviewer 1**

**In this study the biogeochemical and biogeophysical effects of vegetation on the climate system are analysed separately over the time span of the last 120 000 years. Spanning the last ice age inception, last glacial and the latest deglaciation, 62 "snapshot"-type (i.e., under constant forcing) simulations were integrated, distributed over that period, with HadCM3, a fully coupled atmosphere-ocean GCM with interactive vegetation. In addition, 5 transient simulations were integrated over the whole period with various versions of the cGENIE model. These simulations were based on terrestrial carbon fluxes diagnosed from the HadCM3 simulations. The authors conclude that the biogeophysical effects of vegetation account for additional mean cooling during the glacial and, in some cases, substantial regional cooling. The biogeochemical effects are smaller and of opposite sign. The authors also emphasize that different timescales are involved for these two effects on climate.**

**I have enjoyed reading the paper. It is well written and presents interesting results. I do have three major comments and a long list of other comments and recommend that this paper might be suitable for publication after all of my comments are addressed (i.e. major revisions).**

We're very glad the reviewer enjoyed the paper and think it is interesting, and thank them for their helpful and detailed comments. Our responses to the reviewer's comments (in bold) are below (in standard text).

**Major comments:**
**1. One underlying hypothesis with this approach is that the feedbacks considered (ocean, vegetation, atmosphere, sediments, biogeophysical, biogeochemical) add up linearly (i.e. there is no non-linear interaction between the biogeophysical and biogeochemical feedbacks). This assumption needs to be clearly stated and discussed.**

We've added the following to the discussion:

Our approach here assumes that there is no non-linear interaction between the biogeochemical and biogeophysical effects. Since the biogeochemistry acts as a negative feedback and reduces over time, and the biogeophysics acts as a positive feedback and stays the same over time, there's no strong reason to believe that in equilibrium there would be any significant synergy. However, on shorter timescales and on a regional rather than global scale, it is quite possible that there could be some synergies.

**2. Under which boundary conditions cGENIE was spun up? Am I right to assume that 120kaBP boundary conditions were used? Am I also right to assume that the diagnosed terrestrial carbon fluxes (calculated based on changes in terrestrial carbon) from the HadCM3 snapshot simulations were interpolated as "emission" forcing time series and applied to cGENIE? And that these "emission" forcing time series were the only changing boundary conditions during the simulations (i.e., no additional imposed CO2 changes, no continental ice sheet changes, etc)? If I understood this right, there is another assumption made by this approach: that the climate sensitivity is constant and independent of the climate state. This assumption also needs to be clearly stated and discussed.**

All good questions.

Firstly, although the model configuration used was cited, without tediously reading the paper concerned, we admit that it is not clear what specific boundary conditions are assumed here,

particularly as our study considered changes occurring over a full glacial cycle. The cGENIE model configuration was modern (pre-industrial, a.k.a. late Holocene). We have now clarified this explicitly in the text.

So the overarching question (encompassing the reviewer's comments) then arises: what is the sensitivity of the (marine) carbon cycle and climate to an impulse of $CO_2$, considering that the boundary conditions of ice sheets, orbits, radiative forcing, sea-level, ocean circulation and chemistry, are all continually changing? This is way beyond what we can explicitly (i.e. mechanistically in the model) address here, hence the simplistic use of the modern configuration of the cGENIE model. However, we should have been more explicit in the text about the implications of this. We have now added new discussion surrounding this issue and in what ways our methodology might induce artifacts in the diagnosed contribution of biogeochemical effects to the overall glacial-interglacial climate change.

Regarding the details of the emissions forcing in cGENIE – yes, the reviewer is correct (in that they are interpolated from the GCM snap-shots). We have now made this clearer in the text.

**3. As shown in Figure 3b and described on Page 8 (lines 11-15), the vegetation changes trigger a change in ocean circulation. While I agree that this is a model dependent result and not part of the core results (although certainly influencing these core results), it would be interesting to see time series of AMOC for the static and dynamic simulations. In case there is a simple feedback that can be easily crystallized (such as the one mentioned in the text), it would also be good to analyse the results further, verify that this is indeed the feedback and play and describe this feedback in more detail.**

We've added the following explanation of the AMOC changes:

Although the biogeophysical changes cause cooling, there are some minima of biogeophysical temperature change seen at 30 ka, 56 ka and 100 ka (Figure 2, filled symbols). These minima have an oceanic source and are caused by vegetation interacting with thermohaline circulation changes. In our new simulations we account for the net transport of water from ocean to the ice sheets by a parameterisation that instantaneously balances any net accumulation of water on ice. This parameterisation results in fresher ocean conditions during times of precession driven N. Hemisphere summer insolation highs (less water is being used to build the ice sheets). The instantaneous nature of the parameterisation is physically unrealistic but reductions in accumulation and an increase in ablation during precession highs has been seen in fully coupled climate-icesheet EMIC simulations (e.g. Ganopolski et al, 2010). During weaker accumulation periods, the parameterisation results in a freshening of ocean surface waters and a reduction in AMOC strength from ~16Sv to 10-12Sv.

Superimposed upon this general behaviour, the addition of interactive vegetation generally does not change the AMOC strength. However, at times of weak AMOC, small changes in runoff and temperature are sufficient to cause some changes in the response. For instance, in the static vegetation simulations there is a relatively weak AMOC in the simulations for 60ka, 58ka, and 56ka. In the interactive vegetation simulations, the weakened AMOC only occurs at 60ka. Thus at 60ka the changes in climate are fairly typical of preceding times but at 58ka and 56ka there is a substantial difference between the static and dynamic vegetation simulations. The cause for this difference is associated with a combination of reduced runoff into the N. Atlantic (principally from changes in land surface in N. America) and colder temperatures, both of which act to stabilise the AMOC in all three periods but it is sufficient to prevent the AMOC weakening in the 58 and 56ka simulations.

This threshold like behaviour of the AMOC is almost certainly highly model dependent and hence the result is not robust.

**Other comments:**
**\* Permafrost and wetlands are not (cannot be) resolved correctly. Both play important roles in terrestrial carbon feedbacks. While the Discussion briefly mentions the lack of permafrost related feedbacks in the simulations (Page 14, lines 16-23), it might be interesting to add a discussion about wetlands (changes in wetlands in the tropics, but also changes from permafrost to wetlands or vice-versa at high latitudes). Can you please broaden this discussion, including some key references, and, if possible, add an uncertainty range to your results in the Discussion section.**

The following has been added to the discussion:

The soil carbon change under ice-sheets between PI and LGM is modelled as ~220 PgC. Extrapolating from the present day underestimates of the model, we could speculate that this might actually be a third too little. If the true value were ~330 PgC, this would make the total C change PI to LGM 550 PgC. This would put the change more in line with some previous estimates. It would affect the global mean annual biogeochemical contribution by ~0.1 K. This would mean the net effect of vegetation was closer to zero, but the biogeophysical effect would still dominate.

However, the exact size of the terrestrial carbon emissions is uncertain. Other carbon stores not accounted for here are potentially important, for example methane during sea level rises or changes to the wetlands in the tropics. Modelling studies that look at wetlands at the LGM suggest that although the wetland area is larger, but the methane emissions are lower compared to modern day (Kaplan, 2002). However, paleohydrological data indicates a drying in the African tropics (Gasse, 2000). Our model does not have a process based permafrost or wetlands component, and therefore the changes in methane are not accounted for. This is a particular limitation when considering the carbon stored in deep permafrost soils in Northern peatlands. Saito et al. (2013) show that, based on the temperature changes, there is a substantial expansion of permafrost area during glacial times but cannot estimate any changes in carbon storage. Zimov et al. (2006, 2009) have argued that permafrost storage could be a major source of carbon through the deglaciation, and Ciais et al. (2012) argue that there was a large extra pool of inert carbon at the LGM. Similarly, Köhler et al. (2014) have argued that large amounts of carbon were locked into permafrost which were then released rapidly at the Bolling-Allerod.

**\* Coarse resolution and numbers of PFTs (page 5, lines 27-32): The representation of vegetation (and therefore associated feedbacks) is crude in TRIFFID (only 5 PFTs, coarse spatial resolution, crude representation of terrestrial nutrient cycles). While this is not any worse than in most other coupled models, the implications should be discussed in more depth in the discussion section.**

We've added the following about the effect of the PFTs to the discussion section:

The impacts are mainly determined by the vegetation shifts the DGVM simulates. Each gridbox has the potential for 5 PFTs, but generally the Lotka-Volterra equations used in TRIFFID mean that the gridbox is dominated by one PFT. The small number that means the range within each PFT is relatively large. Therefore the model probably underestimates the effects of small perturbations in climate, as the large definition of the PFTs allows the PFT to remain the same. Conversely, it makes an abrupt change more likely as the climate tips a girdbox from being predominantly one PFT to being predominantly another. Overall, the model could be slightly underestimating the amount of change in vegetation. However, because of the ratio of the biogeophyiscal to biogeochemical

changes, if the vegetation change is underestimated, the sign of the net effect of the terrestrial biosphere is unlikely to change. Similarly, because on the long time periods involved much of the released carbon is taken up by the ocean, the changes in carbon densities of the vegetation would need to be wrong by a lot to change the overall signal.

**\* It strikes me in Figure 1 that there is almost no change in tropical rain forest cover. Is that realistic?**

We've added the following discussion to section 3.1:

The forest extent in the tropics at the LGM is similar to PI (see SI Figure 9 for shifts in vegetation at 21 ka). This is supported by pollen and other data (Maslin et al., 2012; Anhuf et al., 2006), and modelling (Cowling et al., 2001) which find that the although there is diminished tropical forest, there is still substantial tree cover at the LGM and little sign of widespread grasslands. Because of the PFT (rather than biome) approach of TRIFFID, and the limited number of PFTs, it's difficult to be sure whether trees in the tropics are a tropical rainforest at the LGM, because they equally could be temperate forest.

**Would it be possible to include a validation of these results (present day bias + comparison to pollen data from LGM for example)?**

The following has been added to the paper:

The climate model used in Hoogakker et al. 2016 is HadCM3B-M1 and the climate model used here is HadCM3B-M2.1. The climate between these two is virtually identical. Since the climate is the main aspect which determines the distribution of vegetation in a DGVM, the verification of Hoogakker's work suggests that the distributions found here are also reasonable.

Comparison with the LGM BIOME6000 dataset shows a broad agreement. The model has considerable expansion of grasses in Eurasia where BIOME6000 has grassland and dry shrubland. Broadly speaking, North America shows little change from the mid Holocene to LGM. One key weakness of the model is in western Europe, where BIOME6000 shows grassland and dry shrubland, whereas the model has shrubs and needleleaf trees.

In the present day, TRIFFID does a reasonable job, as detailed in Valdes et al. (2017).

**\* Page 13, lines 5-8: this is an interesting (although neither surprising nor new) result that feeds back into the discussion about climate sensitivity over long timescales. I would suggest adding a paragraph to the discussion about the different timescales involved and the implications on climate sensitivity. It would also be good to remind the reader, which of these feedbacks are usually incorporated in the state-of-the-art IPCC models (i.e. expand the second paragraph in Discussion).**

The following has been added to the relevant paragraph of the discussion:

From a climate sensitivity point of view, this means that on shorter timescales, the effects of dynamic vegetation can cancel each other out. This provides some rationale for the fact that dynamic vegetation has been generally not included in the majority of state-of-the-art earth system models used in CMIP5, as it doesn't significantly affect the climate sensitivity. At longer time scales, it is more important to include dynamic vegetation, as without the positive feedback of the biogeophysical effects, the climate sensitivity would be under estimated.

**\* Table 1: I am certainly misunderstanding something here. . . Why does the vegetation C change differ for all 4 set-ups? Shouldn't vegetation C only be affected by new land (especially because the atmosphere does not "see" the carbon released from under the ice)? In which case the two ELE simulations should be very similar, if not the same (same for the two ELI simulations)?**

Looking again at table 1, it could be clearer. The vegetation C differs in all four scenarios because all four are compared to the PI veg and soil carbon change, and all have a different combination of land. Since they're all compared to PI, where there is vegetation and soil carbon where at the PI there is glaciers, both are included. Critically, areas that at the LGM had ice-sheets, had forests etc. at PI, so there is some vegetation carbon difference. The rounded values for each aspect are on page 11, lines 3-5. To help make this clearer, we've amended these so the exact values as calculated in the model are given.

**\* HadCM3 simulations: it is my understanding that the HadCM3 simulations are run under constant external forcing, initialized from the previous MOSES 1 simulation, then run for 300 years with equilibrium TRIFFID (50 years of TRIFFID for each 5 years of the climate model run), and finally integrated 300 years with dynamic TRIFFID (where TRIFFID is called every 10 days). I am puzzled by this approach – I would assume that TRIFFID is in equilibrium after the first 300 years of "equilibrium setting". I would also assume that atmospheric conditions are fairly close to equilibrium after these first 300 years, so TRIFFID in "equilibrium setting" saw internal forcing with little drift. Why integrate TRIFFID in dynamic setting for the last 300 years, if the forcing that TRIFFID "sees" is more or less constant in a climatological sense and if TRIFFID is already in equilibrium? This does not change the results presented here, it just seems weird (e.g. why not run TRIFFID in equilibrium over the whole 600 years of constant boundary conditions?). See also text on Page 11, lines 15-16 "the model is run for sufficient length of time for the soil and vegetation carbon to reach equilibrium".**

This is an interesting question, and we agree that it's not immediately evident in the paper what the rationale was for our methodology. We hope the following will help clarify.

In equilibrium mode TRIFFID is prone to considerable fluctuations around the 'true' equilibrium determined by the dynamic mode, as the 50 years it runs with can be prone to bias due to inter-annual variability. Therefore, we consider equilibrium mode a tool for spin-up, and wherever possible use dynamic mode in simulations for publication – that's why we don't just run it for 600 years in equilibrium mode. In dynamic mode TRIFFID can take a long time to fully equilibrate the vegetation, and the soil carbon can only be in equilibrium after the vegetation (because of the litter changes). However, these changes are relatively small.

Therefore, you're right that it probably doesn't make that much difference to the final simulation results. But we consider it due diligence, and have reported as clearly and honestly as possible what we have done to make to create these simulations.

**\* Set of simulations with static vegetation based on PI simulation (page 4, lines 11- 14): I assume that the vegetation in this simulation is masked out under continental ice sheets (adjusted for the ice sheet extension of the period of interest). Can you please add this here.**

You're absolutely right that in the Static simulations the land cover is the PI vegetation cover, with the time-appropriate continental ice-sheets imposed. We specify on line 16 that the ice-sheets are the same in both simulations, we think it's clearer to talk about the differences only, and the

similarities in a separate paragraph. Therefore, to help ease any potential confusion, we've added to lines 11 -14 a note to see below for the details of aspects which are the same in the two sets.

**\* Why are the EPICA anomalies halved? Is that to get a representation of global temperature changes (versus changes in local temperature)? Can you please clarify this in the text?**

You're correct, we use the halved EPICA anomalies as an approximation of global temperature changes. We mention this on page 7, line 2-3, but have also added this to the figure caption.

**\* I do not understand what is shown in Figure 4a. Is it the globally integrated (or globally mean) albedo anomalies? If this is the case; why don't they add up? Or are these means over land/ocean versus global mean? Can you please clarify?**

We apologise, Figure 4 had the incorrect key. We've rectified this, and hopefully it now makes more sense.

**\* Figure 5: can you please show these plots for simulations that treat carbon under ice sheets the same way? These plots should only show the difference in C and albedo due to changes in vegetation and soils away from ice sheets. The large purple areas in 5e and f take the attention away from the results.**

We've changed these plots to mask over the carbon under ice-sheets, and added the original plots to the supplementary information.

**\* cGENIE simulation TRCE needs to be described in more detail in Section 2.2.**

The TRCE is currently explained on lines 26 – 29 and we acknowledge that this could lead to readers thinking it is a cGENIE simulation. To remedy this, we have included the explanation about TRCE in the overview of the methodology at the top of section 2.

**Editorial changes:**
**\* Page 4, line 6: I would prefer to go forward in time: 120 – 0 kaBP.**
Done

**\* Page 4, line 21-24: again, please go forward in time; e.g. 120 to 80 kaBP,. . .**
Done

**\* The filled points are hard to see in Figure 2a**

Whilst we appreciate the filled points are difficult to see in this particular plot, the x axis is the same as on figure 2b, and so the filled points should be easy to infer. Also, the filled points are discussed in relation to figure 2b. 2a is already a 'busy' plot, so we're reluctant to make the points larger, or to break with the pattern set of using the filled points to indicate the time periods of interest.

**\* Figure 4: I find it hard to discern the different lines – would it be a good idea to plot these time series in different colours?**

We've put the lines in different shades of red, so hopefully it's easier to see which are which.  We plotted these line in red to tie in with the colour scheme in other plots, as they all use the GCI-ELE scenario. We're reluctant to use a third colour scheme in the paper, as we feel it could be confusing to readers and reduce the visual cohesiveness of the paper.

**\* Page 9, line 6: "is" missing**
Done.

**\* Figure 6: the red and pink lines are hard to discern**

We've replaced the pink with a green.

**\* Figure 7: Figure caption should read GCI_ELE and not GLI_ELE**

Corrected.

**Comment on Colin's comment: I agree with the authors that this is a sensitivity study, showing the whole range of possible scenarios (including unlikely ones). I would not recommend reducing the numbers of scenarios shown and discussed in the text. However, I agree with Colin that it should be made clear in the abstract that the higher terrestrial carbon loss scenarios are more likely (this is the one the cGENIE time series are shown for in Figure 7).**

We have added to following proviso in the abstract:

In addition, depending on the assumptions about soil carbon under ice-sheets and sea level rise, we find a range in terrestrial carbon storage change from a reduction in LGM carbon storage of -440 PgC, to a gain of +37 PgC, *though we consider the negative part of the range more likely*.

**Side note: a newer estimate of total terrestrial carbon glacial/interglacial change based on benthic d13C data is given by Menviel et al. (2017).**

Thank you for drawing our attention to this paper. We have included it in the discussion.

Reviewer 2

**General comments**

**In this paper the authors present results of 62 equilibrium simulations with a coupled atmosphere-ocean-vegetation general circulation model covering the last glacialinterglacial cycle. They performed simulations with and without dynamic vegetation to quantify the effect of the terrestrial biosphere on glacial-interglacial climate variability in terms of both biogeophysical and biogeochemical effects. Although the results represent an important contribution to our understanding of the role of the terrestrial biosphere to glacial-interglacial variability, I have some major comments that should be addressed before this paper can be considered for publication in Climate of the Past.**

We thank the reviewer for their view that this work is an important contribution to the literature. We address their comments (in bold) in standard font below.

**Since changes in vegetation cover are key to the results presented in this study, the modelled vegetation should be compared with available reconstructions, where available. The BIOME6000 dataset for the last glacial maximum provides a unique reconstruction of vegetation cover for glacial climate conditions and could be used to evaluate the model performance. As a minimum requirement, model results and reconstructions should at least be compared qualitatively and discrepancies discussed. Comparison with other modelling studies would also be useful (e.g. (Prentice et al. 2011; Hoogakker et al. 2016)).**

The following has been added to the paper:

The climate model used in Hoogakker et al. 2016 is HadCM3B-M1 and the climate model used here is HadCM3B-M2.1. The climate between these two is virtually identical. Since the climate is the main aspect which determines the distribution of vegetation in a DGVM, the verification of Hoogakker's work suggests that the distributions found here are also reasonable.

Comparison with the LGM BIOME6000 dataset shows a broad agreement. The model has considerable expansion of grasses in Eurasia where BIOME6000 has grassland and dry shrubland. Broadly speaking, North America shows little change from the mid Holocene to LGM. One key weakness of the model is in western Europe, where BIOME6000 shows grassland and dry shrubland, whereas the model has shrubs and needleleaf trees. Similarly, assessment of the PI vegetation cover of HadCM3 by Valdes et al. (2017) shows good agreement with reconstructions of 1800 vegetation.

**The representation of vegetation cover in Figure 1 in terms of dominant PFTs can be misleading. For example it seems from Figure 1 that tropical forest remains practically unchanged during glacial conditions. Is this just an artifact of the dominant PFT representation or is it a real feature of the model (in which case the discrepancy with available reconstructions has to be discussed)? In any case I would instead suggest showing fractions of all 5 PFTs separately, at least for pre-industrial and LGM.**

We've included the PFT changes PI-LGM separately in the supplementary information. The extent of change of broadleaf forest in the tropics changes relatively little. However, as discussed above, this is similar to many other estimates.

Figure 1 shows the dominant PFT, which inevitably 'hides' not only decrease in the proportion of broadleaf trees, (the dominant PFT is simply the one with the largest proportion, even if that

proportion is low), but also shifts within that climatic envelope. In particular, a broadleaf tree is not necessarily 'tropical rain forest', but equally can be a temperate broadleaf forest, or even savannah-type trees.

We've also added the following discussion to section 3.1:

The forest extent in the tropics at the LGM is similar to PI (see SI Figure 9 for shifts in vegetation at 21 ka). This is supported by pollen and other data (Maslin et al., 2012; Anhuf et al., 2006), and modelling (Cowling et al., 2001) which find that the although there is diminished tropical forest, there is still substantial tree cover at the LGM and little sign of widespread grasslands. Because of the PFT (rather than biome) approach of TRIFFID, and the limited number of PFTs, it's difficult to be sure whether trees in the tropics are a tropical rainforest at the LGM, because they equally could be temperate forest.

**The results about the biogeophysical feedback are presented in a rather superficial way, which makes it difficult to get a quantitative understanding of the processes responsible for the positive feedback. As the authors mention in the introduction, the biogeophysical feedback results primarily from the vegetation controlling the surface energy fluxes. The results section in the paper focuses almost exclusively on the effect of changes in surface albedo. Latent heat effects are only mentioned once when referring to Figure 4b but are not discussed further, and sensible heat flux changes are not mentioned at all. Albedo changes are probably the dominant effect, but the other effects should also be quantified. I would suggest to add panels showing the changes in latent and sensible heat flux to Figure 3 and to move the albedo plots from Figure 5 to Figure 3 (it is not clear to me why albedo maps are shown together with vegetation and soil carbon). Also, I would suggest replacing the albedo figures with net shortwave radiation absorbed at the surface. Shortwave radiation is a more appropriate measure because it accounts for changes in insolation and can moreover directly be compared to the latent and sensible heat fluxes.**

We have added figures of the net shortwave radiation, sensible heat, and latent heat in the supplementary information. Whilst we understand the reasons for the reviewer wishing to see figures of net shortwave radiation absorbed at the surface, we hope that by showing these other metrics in the SI, you'll understand why we feel the albedo is more informative, as well as a more generally understood and used metric.

**The authors show that the vegetation is interacting with the thermohaline circulation. It would be interesting to understand how this is happening. Can anything be said about possible causal relations, given the available simulations? Is vegetation affecting runoff into the North Atlantic, or are vegetation and THC interacting via changes in atmospheric circulation?**

We've added the following explanation of the AMOC changes:

Although the biogeophysical changes cause cooling, there are some minima of biogeophysical temperature change seen at 30 ka, 56 ka and 100 ka (Figure 2, filled symbols). These minima have an oceanic source and are caused by vegetation interacting with thermohaline circulation changes. In our new simulations we account for the net transport of water from ocean to the ice sheets by a parameterisation that instantaneously balances any net accumulation of water on ice. This parameterisation results in fresher ocean conditions during times of precession driven N. Hemisphere summer insolation highs (less water is being used to build the ice sheets). The instantaneous nature of the parameterisation is physically unrealistic but reductions in accumulation and an increase in ablation during precession highs has been seen in fully coupled climate-icesheet EMIC simulations (e.g. Ganopolski et al, 2010). During weaker accumulation periods, the

parameterisation results in a freshening of ocean surface waters and a reduction in AMOC strength from ~16Sv to 10-12Sv.

Superimposed upon this general behaviour, the addition of interactive vegetation generally does not change the AMOC strength. However, at times of weak AMOC, small changes in runoff and temperature are sufficient to cause some changes in the response. For instance, in the static vegetation simulations there is a relatively weak AMOC in the simulations for 60ka, 58ka, and 56ka. In the interactive vegetation simulations, the weakened AMOC only occurs at 60ka. Thus at 60ka the changes in climate are fairly typical of preceding times but at 58ka and 56ka there is a substantial difference between the static and dynamic vegetation simulations. The cause for this difference is associated with a combination of reduced runoff into the N. Atlantic (principally from changes in land surface in N. America) and colder temperatures, both of which act to stabilise the AMOC in all three periods but it is sufficient to prevent the AMOC weakening in the 58 and 56ka simulations. This threshold like behaviour of the AMOC is almost certainly highly model dependent and hence the result is not robust.

**In the model description section, no information is given on the soil carbon representation in the model.**

The following has been added to the model description:

The soil carbon is a single pool, increased by litterfall and decreased by respiration. The soil respiration is controlled by moisture and temperature and returns carbon dioxide to the atmosphere unless, as is the case here, the atmospheric carbon dioxide is fixed. The litterfall is an area-weight sum of the litterfall of the five PFTs in each gridcell.

**I expect the results of the biogeochemical part to strongly depend on how soil carbon is represented in the model. In particular, a proper representation of carbon stored in permafrost is probably crucial to model land carbon storage during glacial times. In the discussion section the authors mention that the model does not have a process-based permafrost component, but this should also clearly be stated in the description section.**

We have added a statement to the model description that there is no permafrost component.

**The amount of carbon which can potentially be buried below the ice sheets will strongly depend on how carbon in frozen soils is treated. The authors should discuss this in more detail. How does the carbon stored in permafrost in the model compare to observational estimates (Hugelius et al. 2014) for the present day?**

The following has been added to the discussion:

Carbon in frozen soils is treated essentially the same way as any other soil. The soil carbon respiration is dependent on temperature and soil moisture. Where the temperature is near or below freezing, little or no respiration occurs, causing a build up or retention of soil carbon. There is no soil carbon in the model under icesheets, therefore the values are extrapolated from the soil carbon present when there aren't icesheets (e.g. at PI).

For present day, Hugelius et al. 2014 shows around 75 – 100 kg carbon m2 far north Siberia, 20 -40 further south. Far northern Canada is much more heterogeneous, with values from 20 – 150 kg C m2. The modelled PI values are on the low side, and much more homogeneous, around 15 – 20 kg C

m2, but is similar to Hugelius et al. 2014 in that it shows far north America to be less consistent, with some higher areas of 35-40 kgC m2 in the far north. (See the now supplementary figure of the loss of soil carbon under icesheets at LGM.)

What this suggests is that while on the correct order of magnitude, the model has a very modest amount of soil carbon that could be considered permafrost. Therefore, we think it's reasonable to include this low estimate of soil carbon in the uncertainties.

The soil carbon change under icesheets between PI and LGM is modelled as ~220 PgC. Extrapolating from a comparison with Hugelius et al. 2014, we could speculate that this might actually be a third too little. If the true value were ~330 PgC, this would make the total C change PI to LGM 550 PgC. This would put the change more in line with some previous estimates. It would affect the global mean annual biogeochemical contribution by ~0.1 K. This would mean the net effect of vegetation was closer to zero, but the biogeophysical effect would still dominate.

**In the biogeochemistry results section the effect of dynamic vegetation on land carbon storage is not discussed, although the differences in vegetation and soil carbon between static and dynamic simulations at LGM is shown in Figure 5 (the 30 kyr maps of vegetation and soil carbon in Figure 5 seem redundant to me). If I understand correctly, in this section only the dynamic vegetation simulations are discussed. This should be clarified in the text.**

In section 3.3 the Dynamic simulations are discussed in relation to the Static simulations. As mentioned in the methods, the Static simulations have the PI vegetation cover and carbon stores (i.e. there is no terrestrial carbon cycle in the Static simulations). Therefore, we consider discussing the difference between the Static and Dynamic simulations to be discussing the effects of dynamic vegetation.

We include the 30 ka maps to highlight that whereas the albedo shows a very different pattern between 21 ka and 30 ka, the overall pattern of carbon changes remains similar but slightly smaller, which accounts for why the net biogeophysical and biogeochemical effects vary considerably between these two simulations. Since the focus of the paper is understanding the net effects of vegetation, we feel they are informative to include.

**A figure showing the differences in land carbon storage between LGM and present day would be helpful.**

We show the carbon storage at the LGM (and all other time points) in the supplementary information (figure 8 in the discussion manuscript). In addition, the differences in land carbon storage can be seen in Figure 5c and 5e for the vegetation and soil respectively (which together make up the land in this model) carbon maps for the LGM. As explained in the methods, because the Static simulations have the PI vegetation cover, they also have the PI carbon storage. Therefore, for the carbon the PI – LGM anomaly is the same as the Static – Dynamic. We have added this information to the Figure 5 caption.

**Specific comments**

**Page 1, line 7: specify that the 62 simulations are 'equilibrium' simulations.**
Done.

**Page 1, line 18: ocean/atmosphere**

Done.

**Page 3, line 13: vegetation carbon -> land carbon**
Done.

**Page 4, line 14: remove brackets**
Done.

**Page 4, line 29: timer -> time**
Done.

**Figure 1: legend is hard to read**
We've increased the size of the text in the legend.

**Page 6, line 7: -0.91◦C was -0.84◦C in the abstract (if I understand correctly)**

Our apologies, the abstract incorrectly used the LGM rather than the largest value. The abstract has been corrected to -0.91 (as shown in Figure 2b) rather than the LGM value of -0.84.

**Figure 3: Are the figures for annual mean characteristics? Please specify.**

Yes, they are mean annual values. This has been added into the figure caption.

**Figure 4: use of different colors for different lines would improve readability**

We've changed the lines to different shades of red as well as the patterns, so hopefully it's easier to read. We plotted these line in red to tie in with the colour scheme in other plots, as they all use the GCI-ELE scenario. We're reluctant to use a third colour scheme in the paper, as we feel it could be confusing to readers and reduce the visual cohesiveness of the paper.

**Page 9, line 6: it IS unclear**
Added.

**Page 10, line 2: carbon stores changes -> carbon stores**
Done.

**Page 14, lines 8-11: shelf carbon stocks values should be positive**
Done.

**Page 14, lines 13-15: check sentence**
Corrected.

[revised manuscript text omitted]

---

## Author Response (AR2)

**Reviewer 1**

**The authors have responded to all of my major comments and I recommend to accept this paper with minor revisions.**

**I would recommend to carefully read again through the whole manuscript. Some sentences are not grammatically correct and in some cases words are missing (including the abstract). Especially the new sections need to be double-checked.**

**A few examples include: Page 1, line 12; page 6, line 13; page 8, line 11; page 9, lines 3-6 (inconsistency in text plus wrong figure number); page 14, line 2; page 17, lines 3 and 4.**

**In addition, lines 24-35 on page 16 need to be rewritten – e.g. it needs to be stated that the discussion moved on to soil carbon, but also, the whole section seems to have been written in a rush.**

Thank you for your useful copy edits. We have resolved the issues you've raised and re-checked the paper thoroughly.

**I still do not understand why vegetation C is different for the two ELE simulations (same for the ELI simulations, table 1).**

We're sorry this still isn't clear, though we're not sure why or where the misunderstanding has arisen. We speculate that the reviewer might think that these values are the anomaly between Static and Dynamic, then the anomaly between PI and LGM. Actually, they're comparing the Dynamic simulations PI – LGM, but perhaps this wasn't sufficiently clear.

We've added more description to the table caption to help ensure other readers are not similarly confused.

**Review 2**

**I still think that the simulated vegetation cover should be compared more thoroughly with BIOME6000 data for the LGM.**

We're sorry that the comparison with data we added didn't meet the reviewer's expectations. We would like to point out that we did do all the things that the reviewer suggested. We've added the following further comparison with BIOME6000:

"South-east Asia shows continued Warm-temperate, Temperate, and Tropical forest where our model simulated Broadleaf trees, which encompasses all of these biomes. The BIOME6000 reconstructions show around a dozen Tundra points on and near the bering land bridge, and our model simulates this as C3 grasses, which is the closest PFT to Tundra.

Over central Asia our model has extensive areas where the dominant land surface type is bare soil, indicating desert or sparse, dry vegetation. BIOME6000 shows a mixture of desert and dry grass/shrubland, which is generally in keeping with the low productivity, low density vegetation indicated in our model simulation.

"…Conversely, the BIOME6000 data finds that the tropical rainforest area was reduced during the LGM (Pickett et al., 2004; Prentice and Jolly, 2000; Bigelow et al., 2003; Harrison et al., 2001) and grasslands expanded, as do some modelling studies (Martin Calvo and Prentice, 2015; Prentice et al., 2011; Hoogakker et al., 2016). It is interesting to note that in the present day Amazon, BIOME6000 shows 3 points of tropical forest; 2 Savanna, 2 Warm-temperature forest; 2 temperate forest; and 3 dry grass/shrubland at the LGM. In our simulations the dominant PFT of the same area is broadleaf-trees. For comparison, Prentice et al. (2011) using LPX have tropical forest over the same domain. Therefore there is little indication that where TRIFFID may be inconsistent with BIOME6000 that another model is necessarily significantly better.

"Because of the PFT (rather than biome) approach of TRIFFID, and the limited number of PFTs, it's difficult to be sure whether trees in the tropics are a tropical rainforest at the LGM, as there are a number of biomes with significant amounts of trees. Although there is little change in PFT in the tropics at the LGM, on the margins there are reductions of vegetation carbon, suggesting a change in vegetation within the large margins of the PFTs used in this model."

Overall, the key issue here for us is what this paper contributes to current scientific knowledge. We are not suggesting that this paper provides any advancement of DGVM LGM biome/PFT modelling. Nor are we suggesting that using a DGVM at the LGM is novel or noteworthy. We are using a well-used and understood climate model and DGVM, to model a novel concept in paleoclimate (the total climate contribution of vegetation) in simulations of long and high temporal resolution time-period (120 ka).

The amounts of carbon which would be affected by the model underestimating the change from say, tropical forest to savanna, are not sufficient to change the overall message and results in the paper. Therefore, we feel that it's reasonable to focus on whether our DGVM is doing a good enough job for our purposes (to look at the global carbon changes to compare to biogeophysical changes).

**In particular, the following response to one of my comments does not make any sense to me:**
**"The climate model used in Hoogakker et al. 2016 is HadCM3B-M1 and the climate model used here is HadCM3B-M2.1. The climate between these two is virtually identical. Since the climate is the main aspect which determines the distribution of vegetation in a DGVM, the verification of Hoogakker's work suggests that the distributions found here are also reasonable."**

**I don't understand how a validation done for the BIOME model driven by HadCM3B can be used as a replacement for a validation of the simulations presented in this study, which**

**use the TRIFFID dynamic vegetation model coupled to HadCM3B and show a very different response of vegetation in some regions.**

As the reviewer suggested in their original review, we've provided a comparison between our simulations and BIOME6000 Mega-Biomes and pointed out regions where our model doesn't do well, on lines 3 – 8 on page 8. On lines 8 – 13 page 8 we compare our tropical results to other data, again, as suggested by the reviewer.

In their original review, the reviewer requested: "comparison with other modelling studies… e.g. … Hoogakker et al. 2016." We have provided that, as the reviewer has shown above. This is not as an alternative to comparing to BIOME6000, but in addition.

Having already been criticised by Reviewer 1 for "neither surprising nor new" results, we were reluctant to further reiterate a DGVM result using HadCM3B over 120 ka, as they are substantially similar to Hoogakker, as we state. We apologise if this isn't clear or helpful. It was an attempt to explain why these DGVM PFT distribution results are not particularly original.

Thus, we're between a rock and a hard place: if we provide the extensive detail one reviewer wants, we will inevitably be accused of being unoriginal by another reviewer, and also dilute the novelty and interest to a general reader. The text we've provided gives verification of the simulations overall and points out the areas of weakness in our model.

**Regarding the tropics, the results presented in this paper show almost no change in vegetation cover between preindustrial and the LGM (Fig. 10). There is definitely evidence from the BIOME6000 dataset that the tropical rainforest area was reduced during LGM and was replaced by savannah/shrubland/grassland (e.g. Figure 3 in Prentice et al, 2011). This seems to be confirmed by at least some modelling studies (e.g. Hoogakker et al., 2016; Martin Calvo and Prentice, 2015; Prentice et al., 2011)). On the other hand a recent study suggests that some parts of the Amazon rainforest where resilient to reductions in precipitation during LGM (Wang et al., 2017). The results presented by the authors should be compared also to these data and critically discussed.**

The reviewer is absolutely right, the results do show almost no change in vegetation cover in the tropics PI to LGM. We discuss this on page 8, line 9.

We have incorporated the references the reviewer has suggested into the discussion on page 8, as shown above. We agree that the more balanced view this provides is beneficial to the paper. We thank the reviewer for drawing our attention to these extra references.

**The authors also write:**
**"In particular, a broadleaf tree is not necessarily 'tropical rain forest', but equally can be a temperate broadleaf forest, or even savannah type trees."**
**I don't think that a grid cell which is covered predominantly by broadleaf trees can be considered to be savannah.**

We're sorry the reviewer seems to have misconstrued our comment as a definite assessment of a region, as we were trying to explain the foibles of our PFT model. We doubt whether it's helpful in this instance to argue about what sort of tree or biome a particular PFT or combination of PFTs is.

**Figure 3. surface albedo is mentioned in the caption, but is not plotted in this figure.**

Thank you for drawing this to our attention; we've altered it to say temperature.

**References**
**Hoogakker, B. A. A., Smith, R. S., Singarayer, J. S., Marchant, R., Prentice, I. C., Allen, J. R. M., Anderson, R. S., Bhagwat, S. A., Behling, H., Borisova, O., Bush, M., Correa-Metrio, A., de Vernal, A., Finch, J. M., Fréchette, B., Lozano-Garcia, S., Gosling, W. D., Granoszewski, W., Grimm, E. C., Grüger, E., Hanselman, J., Harrison, S. P., Hill, T. R., Huntley, B., Jiménez-Moreno, G., Kershaw, P., Ledru, M.-P., Magri, D., McKenzie, M., Müller, U., Nakagawa, T., Novenko, E., Penny, D., Sadori, L., Scott, L., Stevenson, J., Valdes, P. J., Vandergoes, M., Velichko, A., Whitlock, C. and Tzedakis, C.: Terrestrial biosphere changes over the last 120 kyr, Clim. Past, 12(1), 51–73, doi:10.5194/cp-12-51-2016, 2016.**
**Martin Calvo, M. and Prentice, I. C.: Effects of fire and CO2 on biogeography and primary production in glacial and modern climates, New Phytol., 208(3), 987–994, doi:10.1111/nph.13485, 2015.**
**Prentice, I. C., Harrison, S. P. and Bartlein, P. J.: Global vegetation and terrestrial carbon cycle changes after the last ice age., New Phytol., 189(4), 988–998, doi:10.1111/j.1469-8137.2010.03620.x, 2011.**
**Wang, X., Edwards, R. L., Auler, A. S., Cheng, H., Kong, X., Wang, Y., Cruz, F. W., Dorale, J. A. and Chiang, H.: Hydroclimate changes across the Amazon lowlands over the past 45,000 years, Nature, 541(7636), 204–207, doi:10.1038/nature20787, 2017.**